# The Terebelliformia-Recent Developments and Future Directions

Pat Hutchings [1,2,*], Orlemir Carrerette [3], João M. M. Nogueira [4], Stephane Hourdez [5] and Nicolas Lavesque [6]

1 Australian Museum Research Institute, Australian Museum, Sydney, NSW 2010, Australia
2 Biological Sciences, Macquarie University, North Ryde, NSW 2109, Australia
3 Laboratório de Ecologia e Evolução de Mar Profundo, Instituto Oceanográfico, Universidade de São Paulo, Cidade Universitária, São Paulo 05508-090, Brazil; orlemir@gmail.com
4 Laboratório de Poliquetologia, Departamento de Zoologia, Instituto de Biociências, Universidade de São Paulo, Rua do Matão, Travessa 14, n. 101, São Paulo 05508-090, Brazil; nogueira@ib.usp.br
5 CNRS, Sorbonne Université, LECOB, UMR 8222, Observatoire Océanologique de Banyuls, 1 Avenue Pierre Fabre, 66650 Banyuls-sur-Mer, France; hourdez@obs-banyuls.fr
6 CNRS, Université Bordeaux, EPOC, UMR 5805, Station Marine d'Arcachon, 33120 Arcachon, France; nicolas.lavesque@u-bordeaux.fr
* Correspondence: pat.hutchings@australian.museum

**Abstract:** Terebelliformia comprises a large group of sedentary polychaetes which live from the intertidal to the deep sea. The majority live in tubes and are selective deposit feeders. This study synthesises the current knowledge of this group, including their distribution, in the different biogeographic regions. We highlight the new methodologies being used to describe them and the resolution of species complexes occurring in the group. The main aim of this review is to highlight the knowledge gaps and to stimulate research in those directions, which will allow for knowledge of their distribution and abundances to be used by ecologists and managers.

**Keywords:** Annelida; polychaetes; biodiversity assessment; geographical distribution; methods; knowledge gaps

## 1. Introduction

This review of the diversity of the Terebelliformia deals with the taxa previously considered as subfamilies of the Terebellidae Johnston, 1846, namely Polycirridae Malmgren, 1866, Terebellidae Johnston, 1846 (previously referred to as the Amphitritinae) and Thelepodidae Hessle, 1917, together with the closely related family Trichobranchidae Malmgren, 1866, and the recently described family Telothelepodidae Nogueira, Fitzhugh and Hutchings, 2013. For a detailed discussion of the elevation of the subfamilies of the Terebellidae sensu lato (s.l.) to family level, see Nogueira et al. [1] and Hutchings et al. [2]. As well, we include Alvinellidae Desbruyères and Laubier, 1986, Pectinariidae Johnston, 1865 and Ampharetidae Malmgren, 1866, which are all included within the Terebelliformia.

Terebelliformia are common worldwide, including the polar regions, and may be abundant in some areas [3–5]. While some genera are highly speciose, others are represented by few species or only by a single one (for details of genera and numbers of species, see [2] for terebellids, see [6] for pectinariids, see [7] for alvinellids and [8] for ampharetids).

Members of this diverse group are characterised by the presence of multiple grooved buccal tentacles used for selective deposit feeding. Although it is still debatable whether those structures are homologous among all the families of Terebelliformia, we assume they are [1,9] and, therefore, all are of prostomial origin. Due to the extensible characteristic of these structures, they can be easily recognized around their tubes or galleries, rendering these animals the name "spaghetti worms" (Figures 1–3). Typically, the tentacles are smooth, but some polycirrids have papillose tentacles and ampharetids may also have

grooved, smooth or pinnate tentacles. In general, these tentacles are not retractable into the mouth, except in ampharetids and alvinellids, which are able to fully retract them (Figure 1e–g).

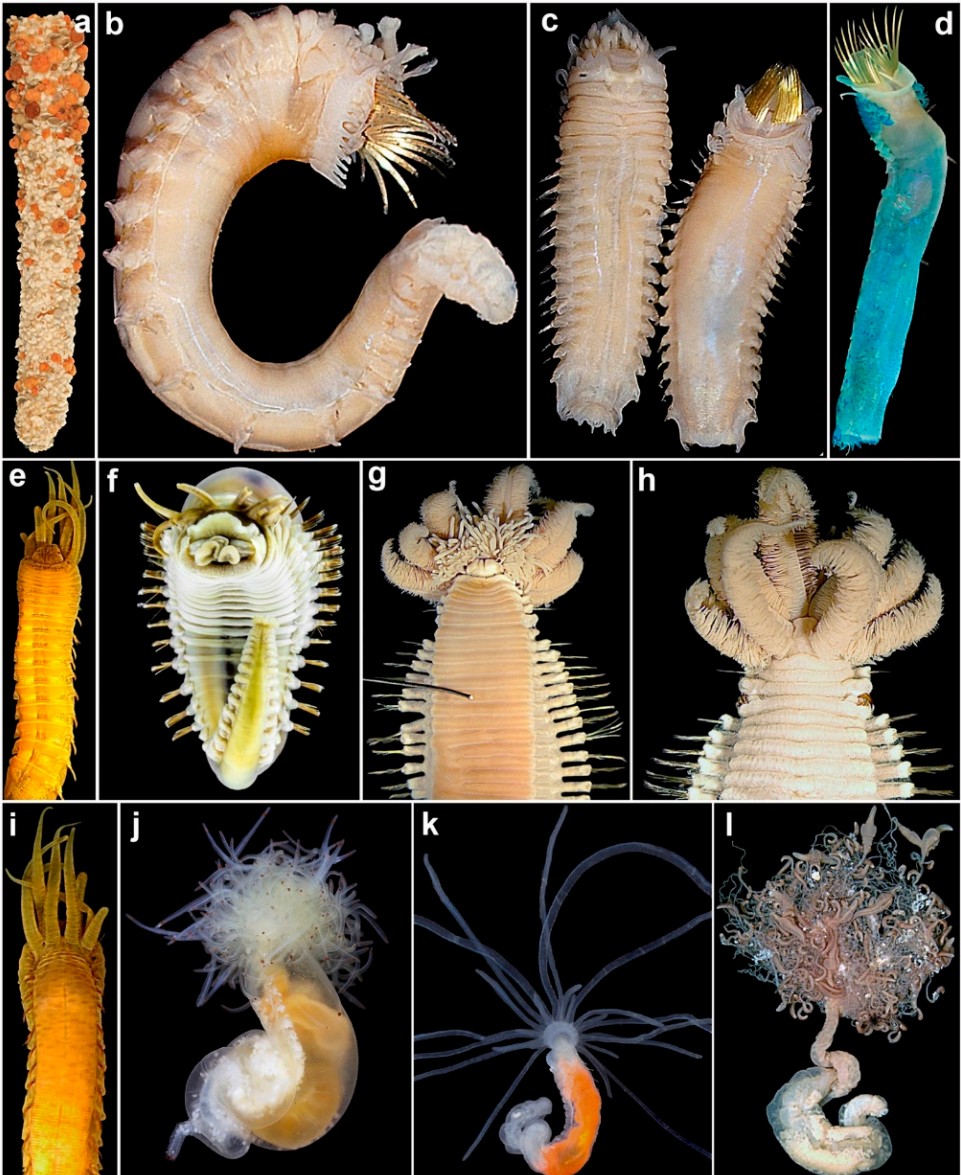

**Figure 1.** Diversity of Terebelliformia: Pectinariidae (PE), Ampharetidae (AM), Alvinellidae (AL) and Polycirridae (PO): (**a**) *Petta investigatoris* (PE), tube; (**b**) *Amphictene auricoma* (PE): entire worm, left lateral view; (**c**) *Petta pusilla* (PE): entire worms, ventral (left) and dorsal (right) views; (**d**) *Petta investigatoris* (PE): entire worm, dorsal view stained in methyl green; (**e**,**i**) *Amphicteis dalmatica* (paratype AM W.11667) (AM): anterior end, ventral and dorsal views, respectively; (**f**) *Amythas membranifera* (AM): entire worm, ventral view; (**g**,**h**) *Alvinella pompejana* (AM W.29585) (AL): anterior end, ventral and dorsal views, respectively; (**j**) *Polycirrus oculeus* (paratype AM W.44612) (PO): entire live worm, dorso-lateral view; (**k**) *Polycirrus rubrointestinalis* (PO): entire worm live, dorsal view; (**l**) *Hauchiella tentaculata* (holotype NTM W.023154) (PO): entire live worm, dorsal ventral view. Photos: (**d**)—E. Wong; (**f**)—Gabriel Monteiro; (**j**–**l**)—A. Semenov.

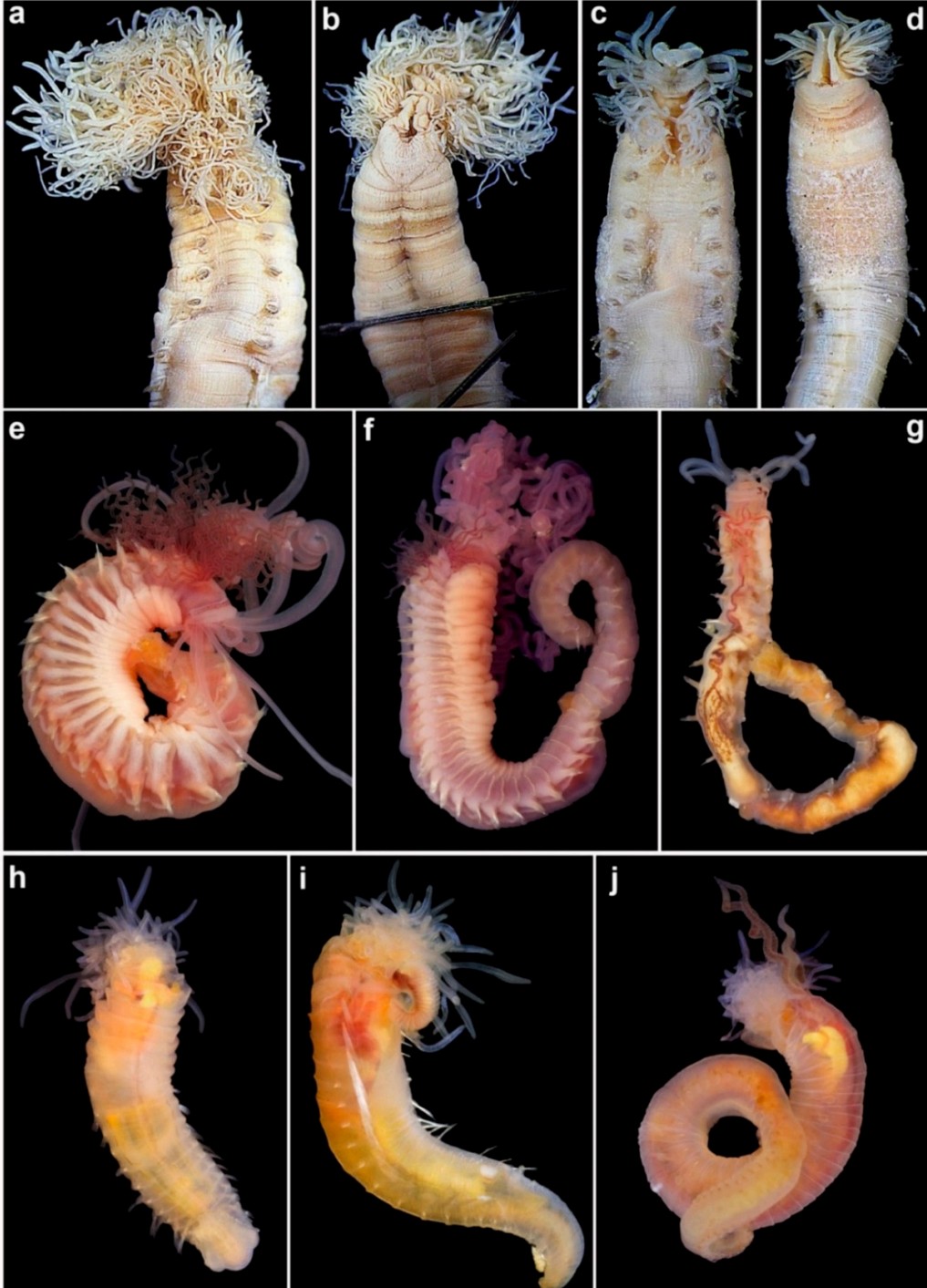

**Figure 2.** Diversity of Terebelliformia: Telothelepodidae (TE), Thelepodidae (TH) and Trichobranchidae (TR): (**a**,**b**) *Telothelepus capensis* (topotype NHMUK ANEA 1955.12.30.1) (TE): anterior end, dorsal and ventral views, respectively; (**c**,**d**) *Rhinothelepus mexicanus* (holotype LACM-AHF Poly 1449) (TE): anterior end, dorsal and ventral views, respectively; (**e**,**f**) *Thelepus paiderotos* (AM W.44600 and AM W.44283, respectively) (TH): entire live worms in right lateral and ventro-lateral views, respectively; (**g**) *Streblosoma curvus* (paratype AM W.44287) (TH): entire live worm (incomplete), dorsal view; (**h**,**i**) *Terebellides akares* (paratype AM W.45450) (TR): ventral and left dorso-lateral views, respectively, of live animals; (**j**) *Trichobranchus hirsutus* (AM W.45444) (TR): complete live worm, left lateral view. Photos: (**e**–**j**)—A. Semenov.

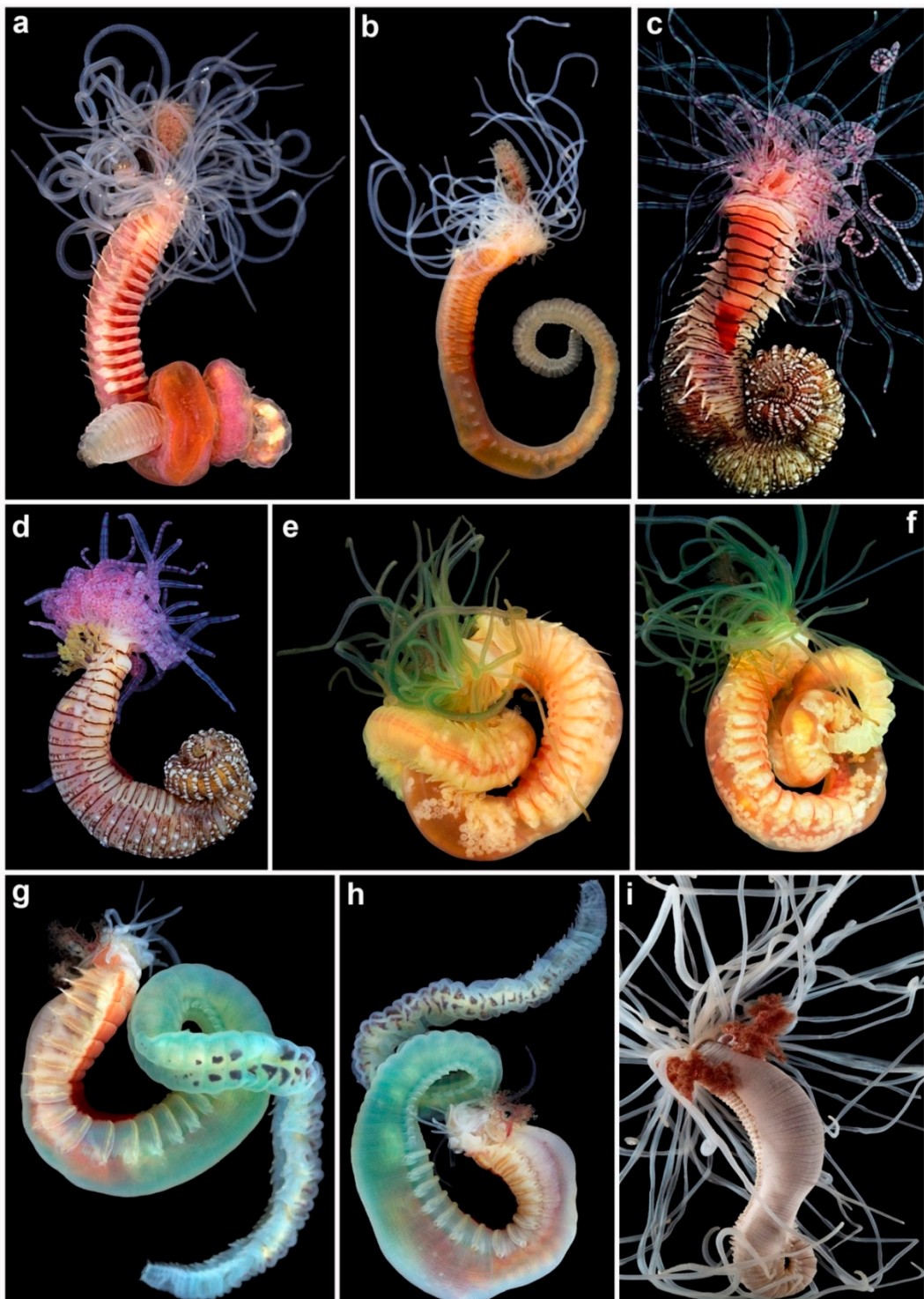

**Figure 3.** (**a**,**b**) Diversity of Terebelliformia: Terebellidae *s.l.* (TER) *Pistella franciscana*: complete live worm, right lateral views; (**c**,**d**) *Loimia tuberculata* (holotype AM W.44280): complete live worm, ventral and right lateral views, respectively; (**e**,**f**) *Pista chloroplokamia* (holotype AM W.44613): entire live worm, female, left and right lateral views; (**g**,**h**) *Loimia pseudotriloba* (holotype AM W.47810): entire live worm, right and left lateral views; (**i**) *Reteterebella lirrf* (paratype AM W.44545): entire live worm, dorso-lateral view. All animals removed from their tubes. Photos: (**a**–**i**)—A. Semenov.

In this paper, we discuss the current status of our knowledge of Terebelliformia, considering all the modern techniques available, which allows for much deeper analyses and observations, including at the molecular level, to document the diversity of the

group. We also discuss the major gaps in our knowledge of Terebelliformia and their phylogeny, including some taxonomic issues, and point to directions to solve them, as well as highlighting other issues which need to be addressed.

The aims of this paper are (1) to present the taxonomic history of these worms, (2) their morphology, (3) the recent studies on their phylogenetic relationships, (4) their roles in the ecosystem and their distribution around the world, (5) the evolution of the methods used to describe them, (6) the knowledge gaps and challenges for the future, with focus on species complexes and taxonomic issues and, finally, (7) how such data can be used in marine park management as well as comments regarding the importance of using correct names.

## 2. Materials and Methods

This study provides a literature review of the Terebelliformia, including a list of valid species and their distribution according to biogeographical regions and their depth ranges (see Supplementary Material). This is based on the literature as well the World Register of Marine Species (WoRMS, http://www.marinespecies.org) to assess the number of currently valid taxa and analyses of species richness.

The citation of authors and date, and type localities policy: the original author(s) and date of a name of all taxa here included are cited the first time they appear in the text. However, due to the large number of taxa cited in this paper, we have not included all these citations in the references. Instead, they can be found in WoRMS as well as details of type localities and synonymies. We also discuss various genera for which diagnostic characters still need to be evaluated. Biodiversity information is referred to the realms proposed by Spalding et al. [10].

## 3. Terebelliforms

### 3.1. Taxonomic History of the Terebelliformia

The discovery of Terebelliformia began in 1766 (Figure 4), with the description of three species from the Dutch Sea, by Pallas: *Lanice conchilega* (Pallas, 1766) (Terebellidae), *Pectinaria belgica* (Pallas, 1766) and *P. capensis* (Pallas, 1766) (Pectinariidae). Since then, more than 1100 species of Terebelliformia have been described by 162 different first authors (Supplementary Material). During this period, four peaks were identified (Figure 4). The initial phase lasted for almost 100 years, from 1766 to 1859, and it was not the most productive, as only 46 species were described. The first peak occurred from 1860 to 1889 when 185 species were described by few taxonomists (Figure 4), as noted by Pamungkas et al. [11]. This productive period can be explained by the publication of important monographs by Europe-based polychaetologists: Grube (47) species) (e.g., [12,13]), Kinberg (12 species) [14], Malmgren (19 species) [15], McIntosh (36 species) [16] and Schmarda (13 species) [17]. Malmgren [15] launched the foundations for the modern taxonomy of Terebelliformia, describing most families of the group and a large number of genera.

By that time, most, if not all, of the researchers were European scientists, working on European material, but frequently no types were deposited in museums or zoological collections, and those species were later often reported from far-away locations. This has led to great taxonomic confusion, which in some cases threatens the stability of important genera (see below). Redescriptions and designation of neotypes from the type localities of some of these early described genera are urgently needed, such as *Amphitrite* O.F. Müller 1771, *Nicolea* Malmgren 1866, *Pista* Malmgren 1866 and *Terebella* Linnaeus 1767, for example.

The second phase of discovery occurred from 1900 to 1919, with 142 new species identified (Figure 4). This period corresponds, once again, to few active taxonomists, such as Augener (12 species) [18], Caullery (8 species) [19], Chamberlin (18 species) [20], Hessle (25 species) [21], Gravier (9 species) (e.g., [22]) and Moore (17 species) (e.g., [23]). It was not until 1970–1989 that the third phase took place, with the description of 165 species, by 34 different first authors. This peak corresponds mainly to the description of new species from Australia by Hutchings and collaborators (59 species) (e.g., [24–30]), but also to the

description of the new family Alvinellidae by Desbruyères and Laubier (12 species, all from deep-sea environments and hydrothermal vents) [31]).

Finally, the years 2000–2019 were the most prolific, with 258 species described by 38 different first authors (Figure 4). Among them, Hutchings, Nogueira and Carrerette were the most productive taxonomists (Table 1), with descriptions of 85 species of Terebellidae s.l., mostly from Brazil and Australia (e.g., [32–40]); Ampharetidae were also well studied during this period, with 32 species described [41–43].

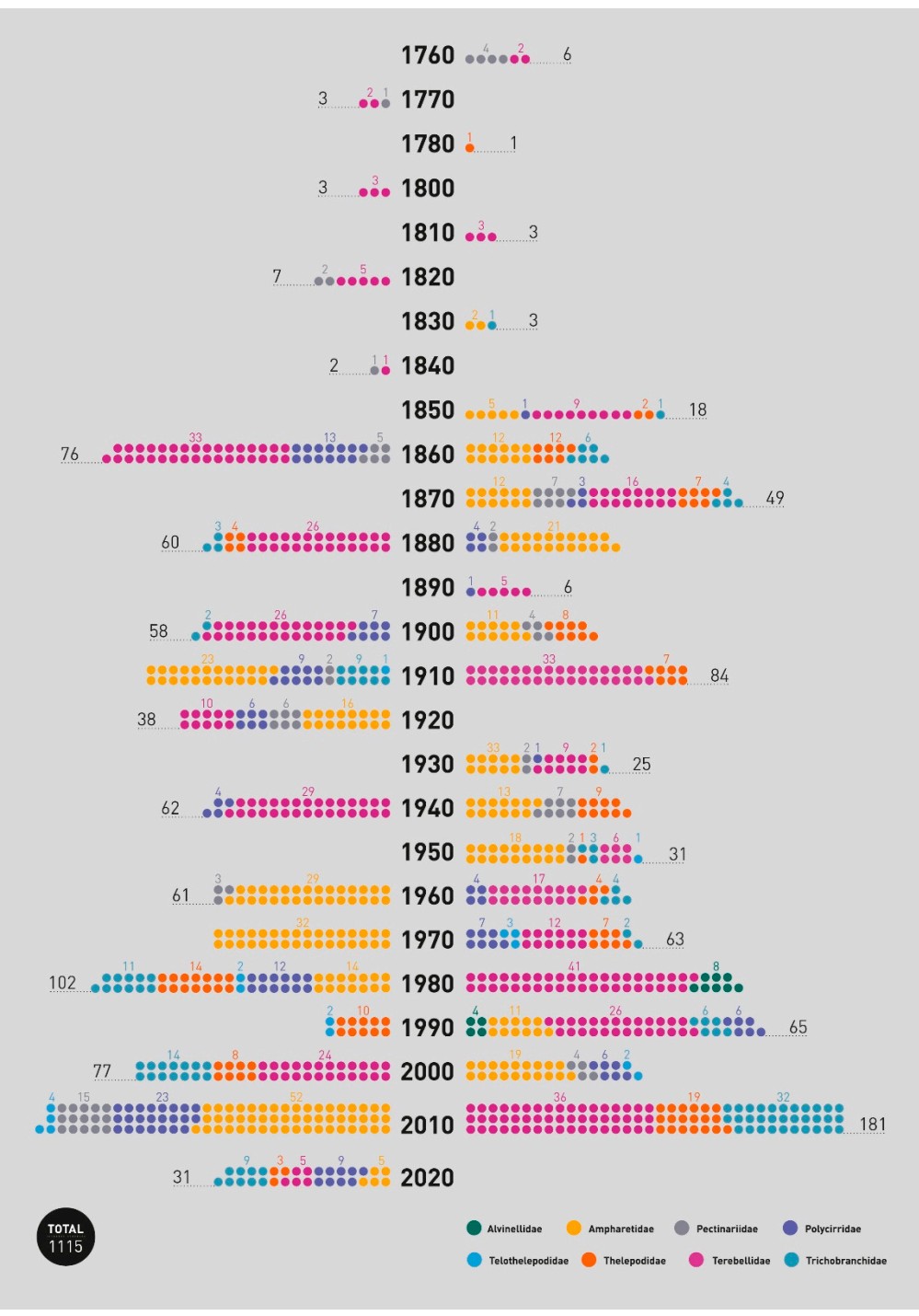

**Figure 4.** Number of Terebelliformia species described per period of ten years.

**Table 1.** The top 30 most prolific authors along with their numbers of Terebelliformia species described, first and last discoveries, and other polychaetes species described. Names in bold refer to active taxonomists.

| Taxonomist | Terebelliformia Species Described | First Record | Last Record | Non Terebelliformia Species Described |
|---|---|---|---|---|
| **P.A. Hutchings** | 217 | 1974 | 2020 | 152 |
| **J.M.N. Nogueira** | 74 | 2010 | 2020 | 54 |
| A.E. Grube | 58 | 1855 | 1878 | 409 |
| M. Caullery | 57 | 1915 | 1944 | 40 |
| **O. Carrerette** | 55 | 2013 | 2020 | 2 |
| **C.J. Glasby** | 49 | 1986 | 2014 | 43 |
| O. Hartman | 45 | 1941 | 1978 | 435 |
| W.C. McIntosh | 43 | 1869 | 1924 | 247 |
| **M.H. Londoño-Mesa** | 38 | 2003 | 2020 | 0 |
| **M. Reuscher** | 33 | 2009 | 2017 | 2 |
| **D. Fiege** | 31 | 2009 | 2016 | 39 |
| G. Hartmann-Schröder | 29 | 1962 | 1992 | 476 |
| **I.A. Jirkov** | 29 | 1985 | 2020 | 11 |
| J.P. Moore | 28 | 1904 | 1923 | 196 |
| C. Hessle | 27 | 1917 | 1917 | 5 |
| M. Imajima | 26 | 1964 | 2015 | 221 |
| K. Fauchald | 25 | 1971 | 1991 | 228 |
| **J. Parapar** | 24 | 1997 | 2020 | 45 |
| **N. Lavesque** | 23 | 2017 | 2020 | 3 |
| R.V. Chamberlin | 21 | 1919 | 1920 | 107 |
| A.J. Malmgren | 21 | 1865 | 1867 | 46 |
| **J. Moreira** | 20 | 2011 | 2020 | 36 |
| J.H. Day | 20 | 1934 | 1973 | 171 |
| A.E. Verrill | 18 | 1873 | 1901 | 102 |
| P. Fauvel | 17 | 1908 | 1959 | 125 |
| H. Augener | 15 | 1906 | 1926 | 197 |
| T. Holthe | 15 | 1985 | 2002 | 1 |
| M. Schüller | 15 | 2008 | 2013 | 8 |
| D. Desbruyères | 14 | 1977 | 1996 | 24 |
| J.G.H. Kinberg | 14 | 1866 | 1867 | 188 |

### 3.2. Morphology of Terebelliforms

Pectinariids are unique among terebelliforms, and among all polychaetes, by having rigid ice-cream cone-shaped tubes [6] (Figure 1a), which disintegrate once the animal dies. These animals are also unique among terebelliforms in having the prostomium and peristomium fused as a cephalic veil, of mixed prostomial and peristomial origin, together with a pair of rows of paleae at the anterior end, and the posterior end modified into a sucker-like scaphe (Figures 1b–d and 5b,i,j,m) [6].

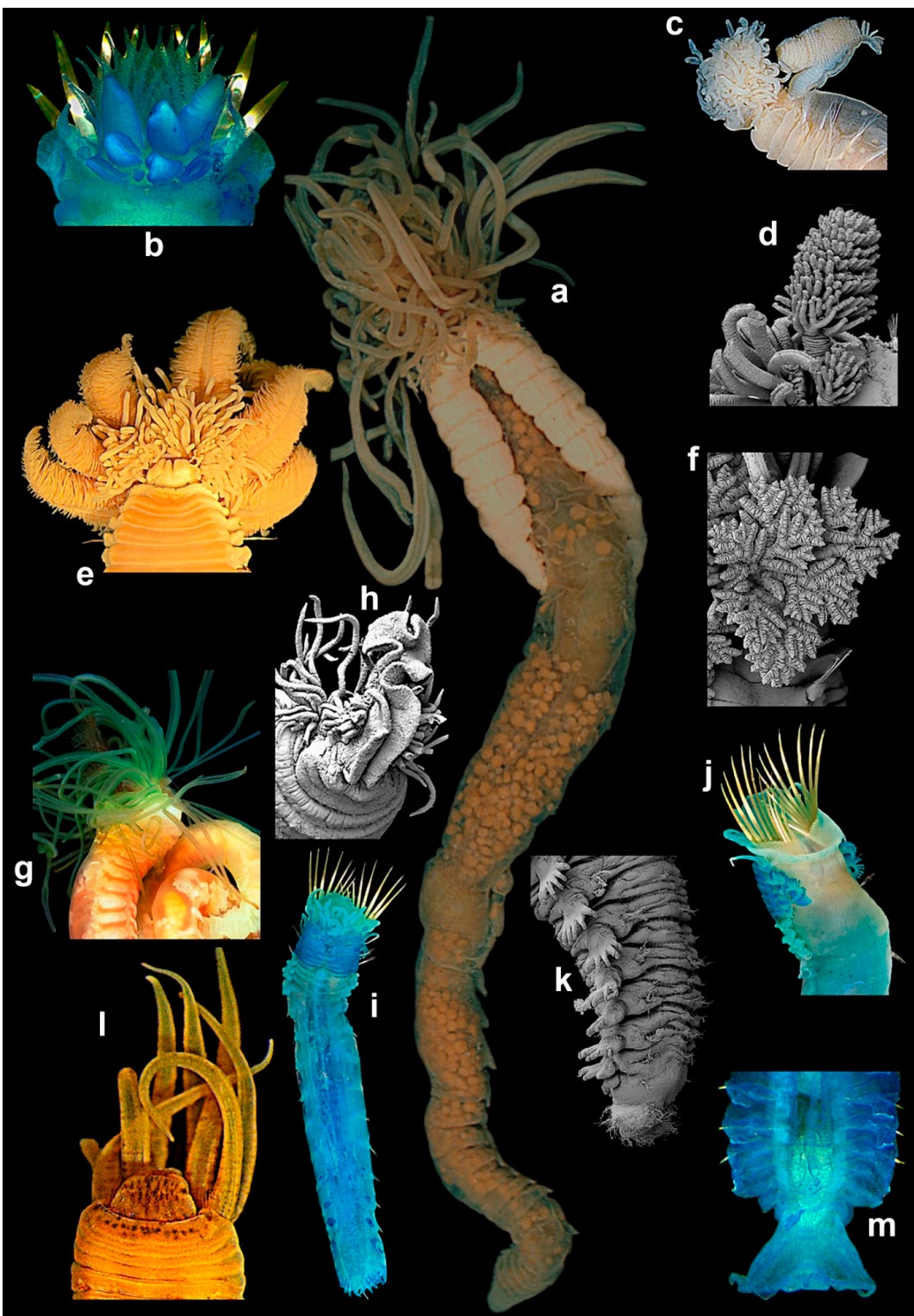

**Figure 5.** Diagnostic characters of terebelliforms: (**a**) *Nicolea lazowasemi* (holotype YPM 40593) (TER): entire worm, a gravid female, dorsal view; (**b**,**m**) *Pectinaria antipoda* (stained in methyl green) (PE): anterior and posterior ends, ventral views, respectively; (**c**) *Terebellides akares* (NTM W.023143) (TR): left lateral view; (**d**) *Pistella franciscana* (paratype AM W.44593) (TER): detail of branchiae (SEM); (**e**) *Alvinella pompejana* (AM W.29585) (AL): anterior end, ventral view; (**f**,**g**) *Pista chloroplokamia* (TER): detail of a branchia (SEM) and anterior end of live animal, right lateral view, respectively; (**h**) *Rhinothelepus occabus* (paratype AM W.201904 (TE): detail of oral area as shown by SEM; (**i**,**j**) *Petta investigatoris* (stained in methyl green) (PE): entire worm, ventral view, and anterior end, left dorso-lateral view, respectively; (**k**) *Trichobranchus hirsutus* (paratype AM W.47510) (TR): posterior end examined under SEM; (**l**) *Amphicteis dalmatica* (AM): anterior end, ventral view.

Alvinellids and ampharetids are more closely related because members of both families have buccal tentacles fully retractable into the mouth and branchiae originating from segments II–V, but arising as free filaments from segments II–III in ampharetids (following Reuscher et al. [44]) [1,8,9,43] and associated to segments III–IV in alvinellids (Figures 1e–i and 5e,l) [7].

In ampharetids, the body regions are well marked, with notopodia restricted to the anterior part of the body (together with neuropodia, frequently called the "thorax"; see below), and posterior abdominal region with neuropodia only (Figure 6l). The shape of the prostomium can vary with the degree of the extension of the tentacles [44,45] but is typically spatulate and swollen, tri-lobed, frequently with paired glandular ridges; these latter structures are also interpreted as nuchal organs [8,43]. Eyespots may be present in ampharetids, and the peristomium is represented by a ring without appendages or chaetae. The first chaetiger is segment II, often with differentiated notochaetae (also referred to as "paleae"), directed upwards (Figures 1e, 5l and 6c), which may be modified or even absent [8,41]. Other thoracic segments usually bear limbate capillary notochaetae (Figure 6d), but some groups present modifications to the anterior segments, including the presence of hook-like chaetae (Melinninae), different sizes and thicknesses of chaetae and notopodia. Notopodia are absent in the abdominal region, although notopodial rudiments may be present (Figure 6p) [9,42,44]. Neuropodia in ampharetids are sessile tori on thoracic segments, forming pinnules after the end of notopodia (Figure 6m); both regions typically bear short uncini, which usually vary in shape and number of teeth between anterior and posterior regions.

In alvinellids, the first chaetae (notopodial only) appear on segment III in *Paralvinella* Desbruyères and Laubier, 1982 and VI in *Alvinella* Desbruyères and Laubier, 1980 (Figure 1g). Neuropodia (uncini) are sessile and start as early as segment VI (chaetiger 4 in *Paralvinella*) but sometimes much later on the body for some species (ca. chaetiger 40). They occur until the end of the body and their morphology does not change markedly in anterior and posterior regions. As a result, body regions are not marked [1,9]. Chaetiger 7 (*Paralvinella*) or 4 and 5 (*Alvinella*) have strong hooks (Figure 1h). Both prostomium and peristomium are devoid of appendages and bear no eyes. All members have four pairs of branchiae, emerging as strong stems bearing lamellae (*Alvinella*) (Figure 1g,h and Figure 5e) or cylindrical extensions (*Paralvinella*). In addition to the typically grooved tentacles, males of alvinellids also possess a pair of short, thick modified tentacles, possibly involved in pseudocopulation.

Terebellidae s.l. is a group of five families previously considered as subfamilies of a single family, Terebellidae ( = Terebellidae s.l.), which Nogueira et al. [1] showed to have originated independently in the evolution of Terebelliformia, raising each of those to family level, and describing a new one, the Telothelepodidae. Animals belonging to these families all have prostomium at the dorsal side of the upper lip, with buccal tentacles originating from the distal part of prostomium, therefore out of the mouth and not retractable into it (Figure 1j,l, Figures 2a–j and 3a–i). In addition, all these animals have up to three pairs of branchiae, usually from segment II, although several forms are abranchiate, including the entire family Polycirridae (Figures 1j–l and 6a); notopodia bearing distally winged (="smooth") (Figure 6d,f,g,i) or serrated capillaries (Figure 6e,h), frequently restricted to the anterior region of the body; neuropodia, extending until near pygidium, bearing uncini (Figure 6j–o,q) [2]. Members of these families, however, are distinguished from each other, mostly by the morphology of the upper lip, the branchiae, the ventral glandular areas of anterior segments, and neuropodia, and by the morphology and arrangement of the uncini of anterior neuropodia, if in single or double rows [2].

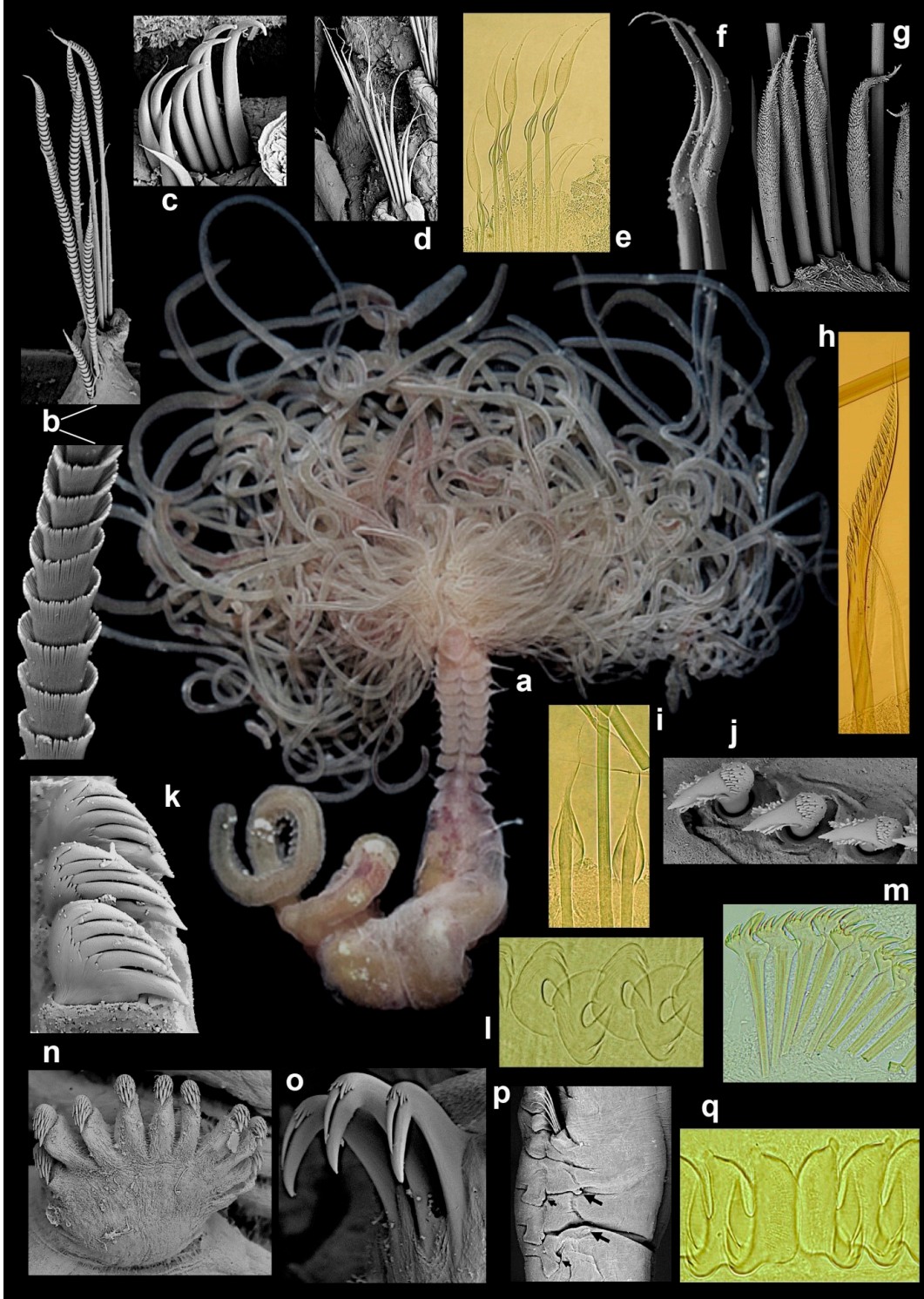

**Figure 6.** Diagnostic characters of terebelliforms: (**a**) *Polycirrus changbunker* (ZUEC 21354) (PO): entire worm, ventral view; (**b**,**o**) *Polycirrus papillatus* (PO): notochaetae, of two magnifications, and abdominal uncini, respectively (SEM); (**c**,**d**) *Amphicteis dalmatica* (AM): paleae and notochaetae of anterior segment, respectively; (**e**) *Spinosphaera barega* (TER): posterior thoracic notochaetae; (**f**) *Pista anneae* (TER): notochaetae, segment X; (**g**) *Pistella franciscana* (TER): notochaetae, segment VIII; (**h**) *Alvinella pompejana* (AL): notochaetae, anterior segment; (**i**) *Leaena ebranchiata* (TER): notochaetae of anterior row, anterior segment; (**j**) *Trichobranchus hirsutus* (TR): uncini, segment VI (SEM); (**k**) *Loimia pseudotriloba* (TER): abdominal uncini (SEM); (**l**) *Nicolea vaili* (TER): uncini, segment 10; (**m**) *Lanicides rubra* (TER): uncini, segment 5; (**n**) *Trichobranchus hirsutus* (TR): neuropodium, segment XXI; (**p**) *Amphicteis dalmatica* (AM): last thoracic and first two abdominal segments; large arrows point to abdominal rudimental notopodia, short arrows point to neuropodial dorsal papillae; (**q**) *Thelepus paiderotos* (TH): uncini, segment VII.

Polycirrids, in addition to being all abranchiate, have a circular upper lip, and the buccal tentacles are of two types, with the long ones often distally modified (Figures 1j–l and 6a). The body may be highly papillated and the anterior glandular areas of anterior segments are typically well developed, with paired mid-ventral shields, separated from each other within pairs by the mid-ventral longitudinal groove, extending from ~segment II or III to the pygidium (Figures 1l and 6a) [1,2,46]. In addition, there is a tendency for a reduction in parapodia in these animals, as members of some genera lack either notopodia (*Biremis* Polloni, Rowe and Teal, 1973), neuropodia (*Enoplobranchus* Verrill, 1879 and *Lysilla* Malmgren, 1866), or lacking all chaetae (*Hauchiella* Levinsen, 1893) (Figure 1l) [2,9].

Members of both Telothelepodidae and Thelepodidae have branchiae as multiple unbranched filaments, originating independently from the body wall on either side of the pairs, 2–3 pairs in thelepodids, on segments II–III or II–IV, always 3 pairs among telothelepodids, on segments II–IV (Figure 2a–c,e–g and Figure 5h). Members of these families are distinguished from each other because telothelepodids have a narrow and proportionally an elongate upper lip, frequently convoluted, very poorly developed ventral glandular areas on anterior segments and distinctly poorly developed neuropodia throughout the body, as low ridges on the anterior body (Figure 2a–d), where notochaetae are also present, and almost sessile pinnules after notopodia terminate. In contrast, members of Thelepodidae have a hood-like, almost circular upper lip and very well developed ventral glandular surfaces of anterior segments, although discrete ventral shields are not observed among these animals; fleshy, well developed neuropodia throughout, the posterior body neuropodial pinnules are frequently well raised from the body (Figure 2e–f) [1,2,9,32,39,40].

Trichobranchids are a group of three genera only, *Octobranchus* Marion and Bobretzki, 1875, *Terebellides* Sars, 1835 and *Trichobranchus* Malmgren, 1866, sharing the character of having neurochaetae on anterior segments as long-handled acicular uncini (Figure 6j), instead of avicular uncini, as in members of all other families, and also poorly developed ventral glandular areas on anterior segments and neuropodia almost sessile on the region with both noto- and neuropodia, and as developed neuropodial pinnules after notopodia terminate (Figures 2h–j and 6n). These animals have a circular, usually flaring upper lip, peristomial lobes are common and the anterior body segments present lobes as low collars of even length around the body, or only ventrally (Figures 2h–j and 5c). An eversible ventral process is present in *Trichobranchus*, in segment 1 [36]. Body regions are well marked in these animals, with notopodia extending only until ~segment XIX or XX, but beginning on segments III–VI, depending on the genus [1,2,9,32]. In *Terebellides*, branchiae are fused into a single structure with two paired lobes that bear lamellae and arise on segments II–IV (Figure 2h,i and Figure 5c) [46]. In contrast, in *Trichobranchus*, branchiae arise from segments II–IV but remain as three pairs of distinct organs (Figure 2j). In *Octobranchus*, there are four pairs present, on segments II–V, which may be digitiform or arranged as a foliaceous structure. Finally, terebellids sensu stricto (s.s.) are unique among Terebellidae s.l. in having neuropodial uncini arranged in double rows on at least some anterior segments (Figure 6l), while animals of all the other families of this group always have uncini in single rows. Terebellids s.s. also have well developed glandular ventral areas of anterior segments, with discrete, unpaired, rectangular to trapezoidal mid-ventral shields, and branchiae, whenever present, originate from a main stalk or at least a single point on the body wall on either side of pairs, and the branchial filaments may be unbranched or branching in a variety of ways (Figures 3a–i and 6a,c,d,f,g) [1,2,9,32,46].

### 3.3. Phylogenetic Relationships within the Group

A detailed discussion on the hypotheses for the position of Terebelliformia within Annelida through time was provided by [2,47,48]. The latest phylogenetic studies, mostly based on molecular data, suggest terebelliforms are a sister taxon to Arenicolidae Johnston, 1835, and the clade is sister to Clitellata, sometimes with Capitellida, Echiurida and Opheliida, also included in the group [49–51]. This contrasts with the traditional morphological hypotheses, which proposed a sister–taxon relationship between Terebelliformia



and Cirratuliformia, grouped together in the taxon Terebellida, which is closely related to Sabellida and Spionida [52,53].

Many of these molecular phylogenies are based on a small number of taxa, and a small number of sequenced genes. Weigert et al. [51], for example, only included two species of alvinellids and one pectinariid, while Zrzavy et al. [50] used one alvinellid, three ampharetids, one pectinariid and two terebellids s.s. This limited number of taxa does not cover the range of morphologies present in the group and often differs from morphological phylogenetic studies. Future studies need a better representation of molecular data from all the families of Terebelliformia, especially of the type species of the genera to continue to resolve the phylogeny of this diverse group of polychaetes.

The most comprehensive phylogenetic study on the phylogenetic relationships within Terebelliformia was performed by Nogueira et al. [1], but was based exclusively on morphological data. The aim of that work was to study the relationships within the Terebellidae s.l., but representatives of the other families of Terebelliformia were also included, as well as three non-terebelliform species, one cirratulid, one spionid and one sabellariid. The authors examined 118 characters in members of 82 species of terebelliforms, including the type species of nearly all genera of Terebellidae s.l., plus the three outgroups, and noticed that all the groups previously considered as subfamilies of Terebellidae had originated independently along the Terebelliformia lineage. As a result, all these groups were raised to the familial level, together with a new family, Telothelepodidae, described therein [1]. According to Nogueira et al. [1], Trichobranchidae is monophyletic, but deeply nested within the Terebellidae s.l., sister to a clade in which Terebellidae s.s. is sister to Alvinellidae/Ampharetidae/Pectinariidae together. All those families originated along the Terebelliformia lineage as follows: Polycirridae (Telothelepodidae (Thelepodidae (Trichobranchidae (Terebellidae s.s. (Alvinellidae (monophyletic Pectinariidae and paraphyletic Ampharetidae))))))). However, the authors stressed that the study was totally focused on Terebellidae s.l., using characters and terminals especially selected for terebellids, but not representative of the diversity of alvinellids, ampharetids and pectinariids; therefore, the relationships between these latter three families had not been properly evaluated [1].

Prior to the study by [1], sister taxa relationships have been suggested between (1) Trichobranchidae and Alvinellidae, the group sister to Pectinariidae, and Ampharetidae and Terebellidae s.l. [54,55]; (2) Alvinellidae and Ampharetidae, and Pectinariidae and Terebellidae s.l., with a plesiomorphic Trichobranchidae, sister to all other terebelliforms [55]; (3) monophyletic Alvinellidae, all other families polyphyletic, except for Trichobranchidae, with a single species included in the study; Pectinariidae is also monophyletic, but out of Terebelliformia [56]. The relationships within Terebellidae s.l. had never been investigated before Nogueira et al. [1], except by Garraffoni and Lana [57,58], who found Trichobranchidae nested within Terebellidae s.l. In their analysis of Terebellidae s.l., Garraffoni and Lana [58] found polycirrids nested within telothelepodids + thelepodids, rendering paraphyletic the traditional Thelepodinae (including species of both thelepodids and telothelepodids, which were regarded as a single family until 2013), and Trichobranchidae sister to Terebellidae.

More recently, a phylogenetic study combining both morphological and molecular data by Stiller et al. [59] suggested a different arrangement for the internal groups of Terebelliformia. The authors first studied transcriptomes of one outgroup plus 20 terebelliform representatives, which included 1 Pectinariidae, 5 Ampharetidae (4 Ampharetinae and 1 Melinninae), 6 Alvinellidae, 2 Trichobranchidae and 6 Terebellidae s.l. (1 Polycirridae, 4 Terebellidae and 1 Thelepodidae), totalling 12,674 orthologous genes, to generate the "backbone" to a more general analysis, with 132 species of terebelliforms (13 Alvinellidae, 49 Ampharetidae (29 Ampharetinae, 5 Amaginae, 8 Amphicteinae and 7 Melinninae), 7 Pectinariidae, 47 Terebellidae s.l. (10 Polycirridae, 27 Terebellidae s.s., 1 Telothelepodidae and 9 Thelepodidae), and 16 Trichobranchidae), combining five genes (three nuclear and two mitochondrial, and not including any of those used for the first analysis) and 90 morphological characters. As a result of the combined analyses, the authors moved the newly

erected families of Terebellidae by Nogueira et al. [1] back into the Terebellidae s.l., most of them as subfamilies and found a sister taxon relationship between Terebellidae and Melinninae, raising the latter to familial level, and also between the remaining Ampharetidae and Alvinellidae. In regard to the Terebellidae s.l., the authors found Polycirridae nested within Terebellidae s.s., and synonymised Telothelepodidae with Thelepodidae, keeping the subfamily status of Thelepodinae and Terebellinae, and suggesting the subdivision of the latter into four tribes, Lanicini, Polycirridi, Procleini and Terebellini. However, although the sampling for the combined analysis is very comprehensive, the one used for the first analysis, which was used as a "backbone" to direct the second study, only included 20 species, of which pectinariids, melinnins, polycirrids and thelepodids were all represented by a single species each, and no telothelepodids were included. In addition, Fitzhugh [60–62] thoroughly discussed the philosophical issues in comparing phylogenetic hypotheses generated by different datasets of characters, as made by Stiller et al. [59] to combine the "backbone" with the main analysis. Fitzhugh also argued against the combination of morphological and molecular data, as well as against molecular phylogenies per se, also due to philosophical issues [62]. We consider that these major changes still need to be re-evaluated, given that only one species of Melinninae was included and the limited sampling of species of Telothelepodidae and Thelepodidae and the validity of plotting morphological characters onto genetic trees. For those reasons, we prefer to follow herein the classification proposed by Nogueira et al. [1], which was subsequently confirmed in the phylogenetic analyses of Polycirridae [46], and Telothelepodidae [47]. However, this may change as additional species are added to the dataset after sequencing.

Another phylogenetic study on the relationships within Terebellidae s.s. was performed by Jirkov and Leontovich [63], which focused on the animals with large lateral lobes only, which they suggest form a monophyletic clade in the family, although the reasons for this were not given. The authors included 93 taxa with large lateral lobes and a single "outgroup" species without lobes, *Terebella lapidaria* Linnaeus, 1767, the type species of the family. They also considered the presence of short-handled or long-handled anterior uncini as a specific character, rather than generic, as had traditionally been considered. As a result, the authors considered only the following genera with large lateral lobes as valid: *Axionice* Malmgren, 1866, *Lanicides* Hessle, 1917, *Lanicola* Hartmann-Schröder, 1986, *Pista* Malmgren, 1866 and *Scionella* Moore, 1963, and synonymised under *Axionice* the genera *Betapista* Banse, 1980, *Eupistella* Chamberlin, 1919, *Euscione* Chamberlin, 1919, *Lanice* Malmgren, 1866, *Loimia* Malmgren, 1866, and *Paraxionice* Fauchald, 1972. The authors also changed the traditional diagnoses of *Axionice* and *Pista* (see below), but these changes have not been adopted by other workers.

In summary, the phylogenetic relationships within the group are still being debated as well as the boundaries of some genera. Hutchings et al. [2] provide a synthesis of the phylogeny of the group prior to the studies by Nogueira et al. [1].

### 3.4. Biological and Ecological Notes on Terebelliforms

3.4.1. Role of Terebelliforms in the Ecosystem

The majority of terebelliforms are tubiculous, living in robust tubes made of sand and sediment grains, which may be within the sediment or more commonly attached to rocks, algae or shells (Figure 7a–f). A few, such as some polycirrids and some alvinellids, lack tubes, instead covered in a mucous sheath. Pectinariids produce very characteristic cone-shaped tubes, using very well calibrated sediment grains (Figure 1a). The alvinellids build tubes on the walls of the vent chimneys, in basaltic cracks with venting (Figure 7f), or live in mucus sheaths at the base of vestimentiferan tubes. In all cases, once the animal dies, the tubes, which are constantly being maintained, tend to break apart, as the mucus binding the shell fragments and sediment particles degenerates.

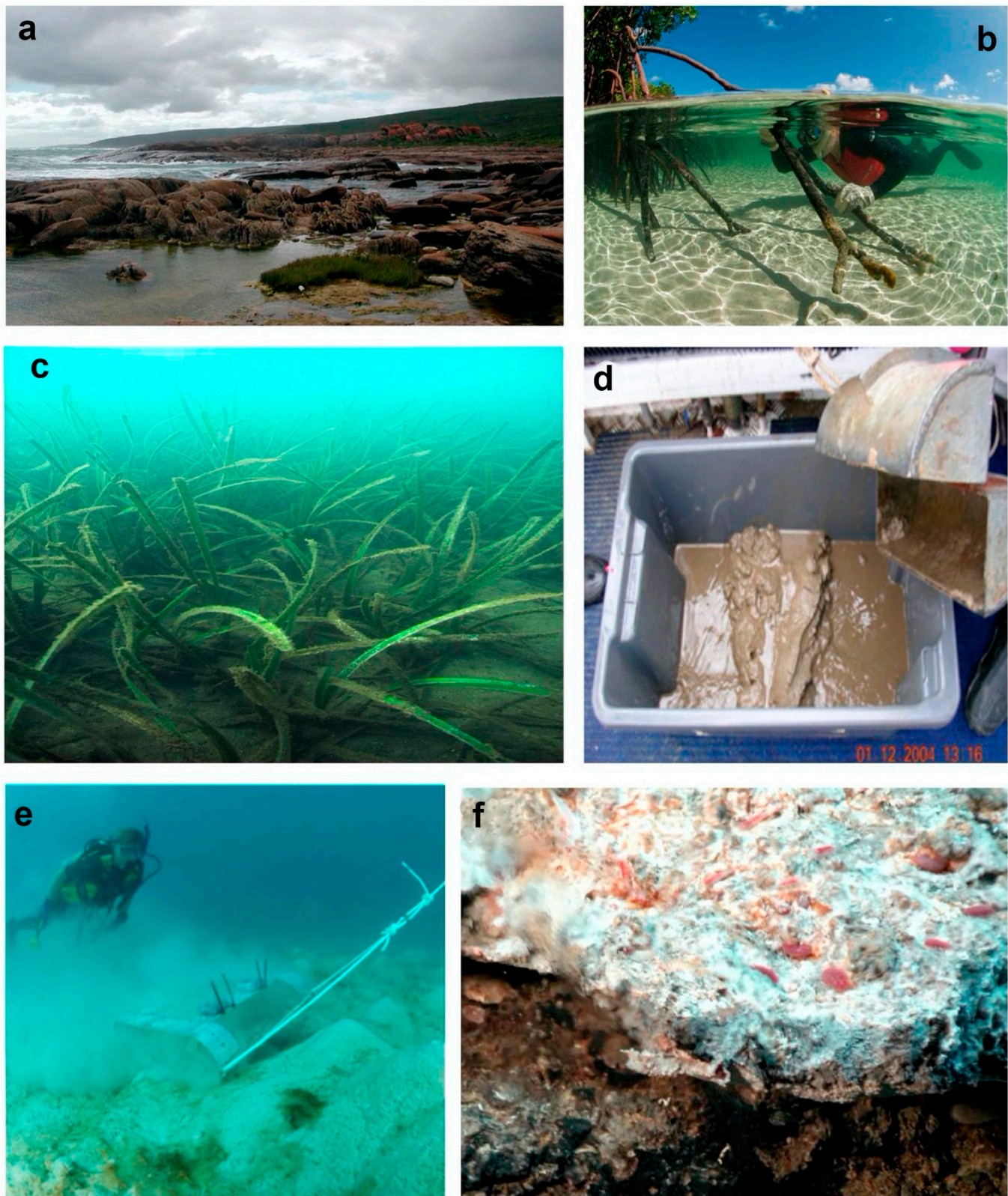

**Figure 7.** Some typical habitats of terebelliforms: (**a**) wave-dominated coastline, Cape Leewin, WA, Australia. Photo: Pat Hutchings; (**b**) mangrove area in front of *Rhizophora* roots at Lizard Island. Photo: Gary Cranitch; (**c**) base of *Posidonia australis* beds. Photo: Clay Bryce; (**d**) soft mud, here collected by Van Veen grab. Photo: Pat Hutchings; (**e**) fine sand sampled by dredge. Photo: Jeurgen Freund; (**f**) deep-sea hydrothermal vents, tubes of alvinellids. Chimney wall surface at Tu'i Malila, Lau Basin. Copyright: Chubacarc cruise/Ifremer.

In general, these animals appear to have reduced mobility; however, members of the polycirrid *Biremis blandi* Polloni, Rowe and Teal, 1973 have been seen swimming in mid-water, at depths of 411–597 m, in the Florida Strait and Bahamas [64]. Other species have been observed to swim for a short time when removed from their tubes, presumably an avoidance reaction (Hutchings, pers. obs.; Nogueira, pers. obs.). On the other hand, species such as *Amphisamytha galapagensis* Zottoli, 1983 can apparently live free of the tubes when the material for their construction is scarce in the environment [65]. They then use fibrous structures such as byssal threads from mussels or setae on crabs to host them.

Some species occur in dense aggregations; for example, the ampharetid *Melinna palmata* Grube, 1870 occurs in aggregations up to 9000 ind./m$^2$ in Arcachon Bay, France [4]. The alvinellid *Alvinella* spp. can also form high-density aggregations on hydrothermal vent chimney walls where it affects the chemical conditions [66] or *Lanice conchilega* (Pallas, 1766), which is also considered as an ecosystem engineer for forming reef-like structures in intertidal sandy substrates, by the aggregation of their tubes [67]. Other terebelliforms, in contrast, form small aggregations or are found as single, solitary individuals.

All the terebelliforms are selective surface deposit feeders [68] gathering food particles with the buccal tentacles, and then conveying these to the mouth, through the ciliated longitudinal tentacular groove. This trophic mode largely modifies marine benthic environments by reworking large amounts of sediments [69] and directly affects their physical and chemical properties [70,71]. Particularly, terebelliforms have a great impact on the amount of organic matter at the water–sediment interface, modifying local hydrodynamics and sediment cohesion [72]. Finally, terebelliforms can influence the structures of benthic communities through tube-building [70].

The Alvinellidae, in addition, have been reported to supplement their deposit feeding diet by collecting particles suspended in the water, by filtering water through their branchiae, as well as feeding on the bacterial residents of the worm tubes [73]. Evidence of such supplementation, however, is lacking and gut contents only revealed mineral particles and bacterial cells gathered from the environment [74]. Although both species of *Alvinella* bear epibiotic bacteria, these do not appear to contribute to the nutrition of the worm hosts. Both species, however, produce structures that allow for the settlement of these bacteria and the association must be beneficial to the host [74].

Most Terebelliformia are dioecious with no morphological differences between males and females, except at the time of spawning when the mature gametes colour the body, where females may be pinkish or greenish, and males are typically cream coloured. In alvinellids, however, reproduction appears continuous; males bear a pair of modified buccal tentacles and females have genital pores [31]. In some taxa, the genital papillae may vary between sexes, as well as the distribution of glandular areas (Figure 6a) [36].

To date, no evidence of asexual reproduction has been observed, although all are capable of regenerating posterior ends, branchiae and buccal tentacles. Gametes are proliferated from the germinal epithelium, often associated with the nephridia, and released into the coelomic cavity, where vitellogenesis and spermatogenesis occur. Synchronised spawning occurs through the nephridia, and spawning varies from only one or two days to discrete periods over several months.

In alvinellids, the presence of sperm ducts, spermathecae and oviducts have been reported, lending support to continuous gamete production, episodic release, pseudocopulation, and internal fertilization [75]. Compared to other studied Terebelliformia, alvinellid sperm cells are highly modified entaquasperm, devoid of acrosome and sometimes of flagella, providing further support for internal fertilization in this family [76].

Among the other terebelliforms, mass spawning occurs in some taxa, while others produce a lecithotrophic larva, with varying planktonic larval phase durations (PLDs), and at least one species has a direct development within a cocoon, with larvae released at the 15 chaetiger stage [74]. Although few species have been studied, most of them produce large yolky eggs, and the embryo probably does not feed in the plankton [8,74], except in pectinariids, which may have a planktotrophic larva, capable of feeding through

a capture system involving the generation of a current and the production of mucus [77]. In *Alvinella pompejana* Desbruyères and Laubier, 1980, the conditions the adults experience in their environment are actually harmful to the developing larvae and they need to encounter milder conditions to survive and develop properly [75]. Erpochaete larvae of *Paralvinella grasslei* Desbruyères and Laubier, 1982 as young as 13 segments (with a single pair of branchiae) have been captured near adults of this species, suggesting a very early recruitment following a planktonic phase [7].

Few species of ampharetids have been studied with regard to their reproduction (e.g., [65,78–85]. Some shallow water species reproduce annually, and all produce large yolky eggs, which spend only a few days in the plankton. *Melinna palmata*, for example, may spend 6 days living in the plankton before settling and building a tube at the 3-chaetiger stage [8]. *Hobsonia florida* (Hartman, 1951) has been reported as having larval development in the maternal tube and a 2-chaetiger stage leaves the tube, settles on the nearby sediment and builds its own tube [8]. Studies conducted with deep-sea species, from both hydrothermal vents and organic falls, suggest continuous reproduction and rapid maturation, possibly as a reflection of the ephemeral conditions of these chemosynthetic habitats [65,84,86].

So, in summary, among terebelliforms, we have a variety of reproductive strategies (see references in [2,7,8]).

### 3.4.2. Distribution and Biogeography

Historically, species were described from Europe with most of the type species of the 137 genera sampled from these waters, except for the Alvinellidae, which is restricted to deep-sea hydrothermal vents, and was erected in 1986 (although the first species was described in 1980, as an aberrant Ampharetidae *Alvinella pompejana*).

In the 1980s, the centre of gravity moved to the southern hemisphere with taxonomists based in Australia (Glasby, Hutchings), South America (Carrerette, Londoño-Mesa, Nogueira) and those involved in Pacific expeditions (Fiege, Imajima, Reuscher) (Table 1). In these regions, polychaete workers had to review earlier expeditions, which were mainly housed by European institutions, where the material was deposited, and over time some has been lost or damaged. In some cases, material from an expedition was deposited in several institutions and locating this material can be challenging. All the scientists working on the material collected during these expeditions were based in Europe and they tended to identify much of this material as European species even though they had been collected thousands of kilometres away in very different habitats and temperature regimes. This led to the idea that many polychaete species were cosmopolitan and certainly later European workers such as Fauvel [86] reinforced this view and recorded the widespread distributions of many species. Later this was reinforced by the catalogues of polychaetes produced by Hartman [87] and by Day [88]. An example of this is provided by Hutchings and Glasby [89] who analysed the species list of terebellids s.l. produced by Day and Hutchings [90] in their checklist of the polychaetes recorded from Australia and New Zealand, which was based entirely on the literature and listed 32 species in 17 genera. Hutchings and Glasby [89] showed that only 14 of these occurred in Australia, the rest having been misidentified as European species. They further analysed the diversity of Australian terebellids, as they were known in 1991, which was represented by 78 species in 27 genera, and of these 67 (85.89%) species were Australian endemics, and of the remaining 11, nine were restricted to the Indo-Pacific and only two were found in the northern hemisphere. Since that analysis in 1991, the remaining two species recorded from the northern hemisphere have been described as Australian endemics as well [91]. Since that study, many more species have been described from Australia. Similar patterns are also evident in other polychaete families, so this is certainly not solely a characteristic of terebellids but widespread across all polychaete families found in Australia.

More recently, it has been recognised that most species actually have discrete distributions, unless proven otherwise [92], and while many genera are widely distributed,

it is at the species level that discrete distributions occur. However, in many parts of the world, taxonomists and ecologists still identify their material using the well-illustrated monographs of Faune de France ([86,93], and Southern Africa [88], despite their samples being collected many thousands of kilometres away from France or Southern Africa. This has tended to reinforce the concept that polychaete species are cosmopolitan in their distribution. While this has been shown not to be true—for example, *Terebellides stroemii* Sars, 1835 is now known to represent a highly speciose group—as the nominal species is restricted to a very small area in Western Norway [5,94]. In many cases, this is also because no regional keys exist in many parts of the world, and so a student has little option but to use keys from other regions. Once those names become enshrined in the local fauna, then subsequent workers just repeat them, explaining why species such as *T. stroemii* have been so widely reported.

Our current knowledge on the diversity of Terebelliformia shows great variation from some regions of the world to others. While places such as Europe and North America have been investigated for centuries, others are still virtually unknown, as is most of the African coast and the Eastern Indo-Pacific. This is largely due to the presence of more researchers based in Europe and North America than in other regions of the world, and also for the availability of financial resources available for biological research in these regions.

However, it should be stressed that even regions where the fauna has been studied, it is now being re-examined with molecular tools, as numerous complexes of species have been found, resulting in a much greater number of species than previously considered. For example, French coastal waters are well-known areas, studied for several centuries by early taxonomists and benthic ecologists. However, studying numerous terebelliforms (spaghetti worms), within the collaborative Spaghetti Project, using modern tools, such as the scanning electron microscope and molecular analysis, has revealed the existence of more than 20 species new for science [95–99]. We assume that this marked discrepancy in our knowledge of the diversity of polychaetes in many parts of the world is common for most if not all polychaete families.

To facilitate a review of the distribution of terebelliforms, we chose to look at various biogeographic schemes which have been suggested over the years ([100,101] and we are following Spalding et al. [10]. In an effort to strategically plan exploitation and marine conservation measures, Spalding et al. [10] suggested a classification for the marine biogeographic regions, the Marine Ecoregions of the World (MEOW), dividing coastal and shelf areas into 12 Realms, 62 Provinces and 232 Ecoregions (Figure 8). As said above, our analysis of the geographic distribution of terebelliforms follows that biogeographic classification. We have compiled a list of all terebelliforms described and, just using their type locality, allocated them to each of these regions and they are plotted in Figure 8. Obviously, these numbers are influenced by the number of taxonomists working in each area, which has varied over time, and the resources available.

As expected, the most diverse realm is the Central Indo-Pacific, with 233 species of terebelliforms described from the region (Table 2, Figure 8). This realm, corresponding to the area from the South China Sea, through the Pacific side of Indochina Peninsula, Philippines, Indonesia, Papua, Melanesia and Micronesia islands, Northwestern, Northern and Northeastern Australia, including the northern Great Barrier Reef (Figure 8), is considered as the world biodiversity hotspot for many groups of marine animals and is referred to as the Coral Triangle [102]. The following most diverse realms match the observations discussed above, as the regions with more polychaete taxonomists and economic resources are also the most diverse, Temperate Northern Atlantic (210 species), Temperate Northern Pacific (175 species), Tropical Atlantic (95 species), Southern Ocean (82 species), Temperate Australasia (76 species) and Temperate South America (67 species) (Table 2 and Figure 8).

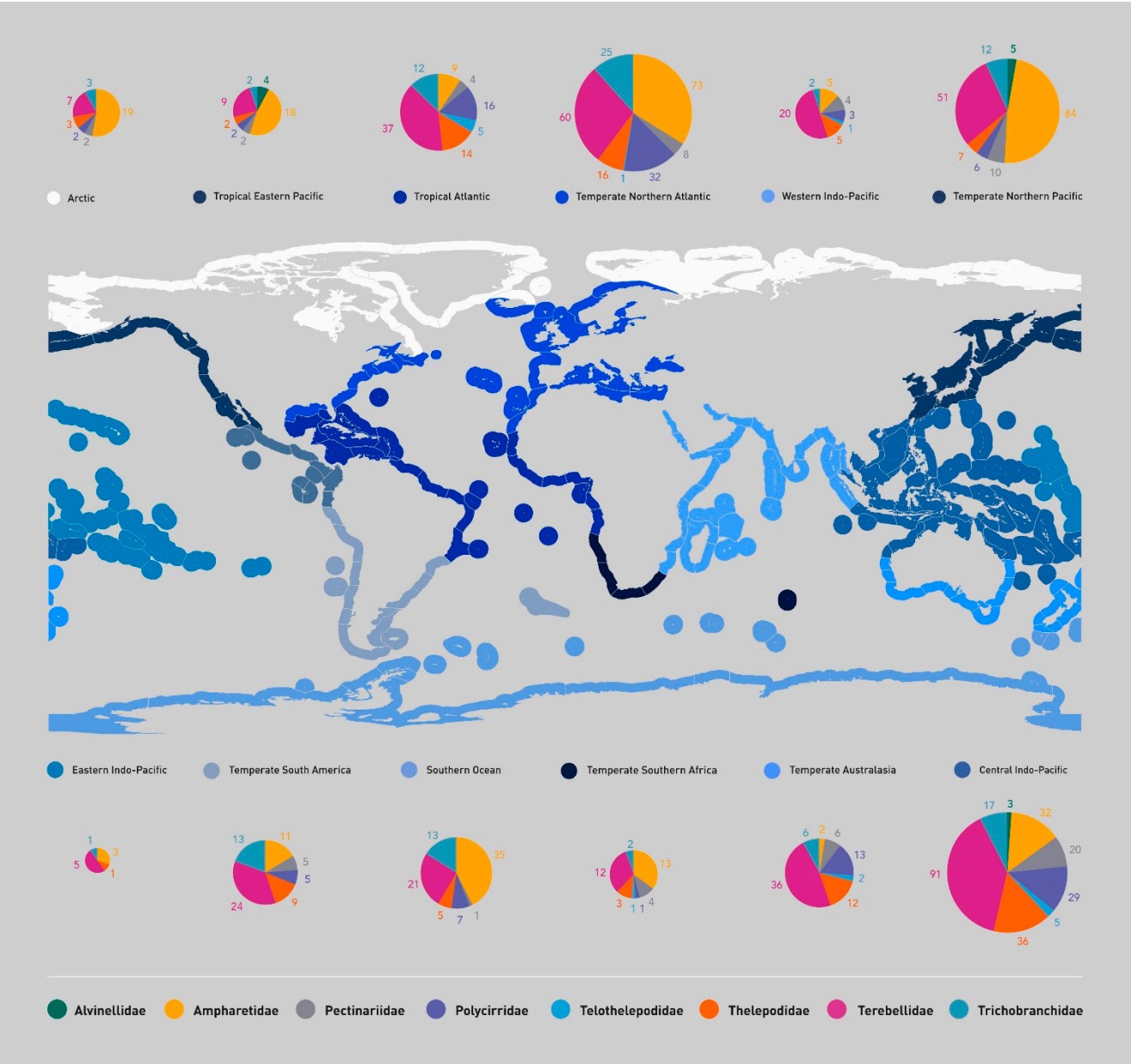

**Figure 8.** Marine Ecoregions of the World following [10] with number of species of terebelliforms described from each realm.

The fauna from Europe, corresponding to part of the Temperate Northern Atlantic realm, has been thoroughly investigated since Linnean times. Despite this long history, many new species are still being found [95–99], as discussed above for French terebelliforms. North America, corresponding to the remainder of the Temperate Northern Atlantic realm, and parts of the Temperate Northern Pacific and Tropical Atlantic realms, certainly had the most taxonomists and economic resources during the 20th century. Temperate Northern Pacific and Tropical Atlantic realms also include some countries which have dedicated resources to intensively study invertebrate taxonomy in the last few decades, such as Russia, Japan and China in the first case, and Mexico, Brazil and Colombia in the latter. The Southern Ocean was investigated by earlier expeditions, but as many countries established scientific bases in Antarctica, this has led to more taxonomic studies. Temperate Australasia and Temperate South America also include countries which have put an effort on the study of marine fauna in the last few decades, with large projects carried out, such as in Australia and New Zealand, in the case of Temperate Australasia, and Chile, Argentina and Brazil, in Temperate South America.

**Table 2.** Distribution of Terebelliformia around the world, following the marine regionalization created by Spalding et al. [10], and bathymetric variation of the terebelliform families, as well as the deepest records of each group.

| | Alvinellidae | Ampharetidae | Pectinariidae | Polycirridae | Telothelepodidae | Thelepodidae | Terebellidae | Trichobranchidae |
|---|---|---|---|---|---|---|---|---|
| **Realms by Spalding et al (2007)** | | | | | | | | |
| Arctic | | 19 | 2 | 2 | | 3 | 7 | 3 |
| Central Indo-Pacific | 3 | 32 | 20 | 29 | 5 | 36 | 91 | 17 |
| Eastern Indo-Pacific | | 3 | | | | 1 | 5 | 1 |
| Suthern Ocean | | 35 | 1 | 7 | | 5 | 21 | 13 |
| Temperate Australasia | | 2 | 6 | 13 | 2 | 12 | 36 | 6 |
| Temperate Northern Atlantic | | 73 | 8 | 32 | 1 | 16 | 60 | 25 |
| Temperate Northern Pacific | 5 | 84 | 10 | 6 | | 7 | 51 | 12 |
| Temperate South America | | 11 | 5 | 5 | | 9 | 24 | 13 |
| Temperate SouthernAfrica | | 13 | 4 | 1 | 1 | 3 | 12 | 2 |
| Tropical Atlantic | | 9 | 4 | 16 | 5 | 14 | 37 | 12 |
| Tropical Eastern Pacific | 4 | 18 | 2 | 2 | | 2 | 9 | 2 |
| Western Indo-Pacific | | 5 | 4 | 3 | 1 | 5 | 20 | 2 |
| **Bathymetric distribution** | | | | | | | | |
| Inter tidal to 100 m | | 72 | 20 | 66 | 14 | 57 | 87 | 34 |
| 100–500 m | | 52 | 5 | 9 | | 6 | 13 | 21 |
| 500–1000 | | 42 | 2 | 3 | | 4 | 1 | 3 |
| 1000–2000 | 4 | 38 | | 1 | | 4 | 1 | 5 |
| 2000–3000 | 7 | 23 | 2 | | | 2 | | 5 |
| 3000–4000 | 1 | 21 | | | | 1 | | |
| 4000–5000 | | 25 | | | | | 1 | 3 |
| 5000–6000 | | 9 | | | | | | 4 |
| 6000–7000 | | 1 | | | | | | |
| 8000–9000 | | 1 | | | | | | |
| 9000–10,000 | | 1 | | | | | | |
| Deeper records | *Alvinella pompejana,* 2593 m | *Anobothrus auriculantus,* 9584 m | *Petta assimilis,* −3000 m | *Polycurrus nonatoi* 1904 m | *Prathelepus anomalus* and *Rhinothelepus mexicanus,* 91 m (for both) | *Streblosoma chilensis,* 3950 m | *Pista torcuata* 4540 m | *Terebellides bulbosa* and *T. ginkgo,* −5200 m (for both) |

On the other hand, the least diverse realms are also those with fewer taxonomists and frequently fewer economic resources. Western Indo-Pacific (including eastern Africa, Red Sea, Persian Gulf, Pakistan, India, Sri Lanka and the Indian side of the Indochina Peninsula) tally only 40 species. Temperate Southern Africa tallies only 37 species, in spite of the efforts by Day [90], but this last author reported many European species for South Africa, as did Fauvel [103], for the region of India and Sri Lanka. Tropical Eastern Pacific, which includes the Pacific side of Tropical America, comprises 38 species only and this is attributable to Mexican and Colombian researchers. Only 36 species are described from the Arctic, which is somewhat surprising, considering the Scandinavian and Russian scientists who have been working in the region since the 19th century, although many Northern European and North American species are reported for this region. Additionally, the Eastern Indo-Pacific realm, the poorest of all, including the region from Hawaii and Marshall Islands through Polynesia and the Mariana Islands to Easter Island, with only 10 species, but also with many records from the West Indo-Pacific (Table 2; Figure 8). We suggest that some of these patterns of diversity may just reflect the lack of sampling rather than a reflection of their true diversity.

As discussed below, many genera of terebelliforms are monotypic, several of which have never been sampled since they were first collected. These descriptions are very brief and do not mention several characters currently considered important for the taxonomy of the group. This is further complicated by the loss of the type of material or it is damaged in such a way that those characters cannot be assessed. The uncertainty about the identity of those genera obviously imposes several problems in regard to the knowledge on the distribution of those animals and several genera are considered as endemic to certain regions which may change as more studies are carried out.

Most of the non-monotypic genera of terebelliforms are widespread through [10] realms. One non-monotypic genus which apparently has a more restricted distribution is *Reteterebella* Hartman, 1963, with three species. The type species, *R. queenslandia* Hartman, 1963, described from Heron Island, Great Barrier Reef (Central Indo-Pacific), but apparently restricted to that region [37], *R. aloba* Hutchings and Glasby, 1988, from South Eastern Australia (Temperate Australasia) and *R. lirrf* Nogueira, Hutchings and Carrerette, 2015, described from Lizard Island, also on the Great Barrier Reef. The habitats in which *R. lirrf*

and *R. queenslandia* occur are very different, the first being found in crevices deep down in boulders, whereas *R. queenslandia* occurs on reef flat with its flimsy tube attached to the underside of boulders. However, it should be stressed that reefs between Heron and Lizard have not been well sampled.

Another genus *Hadrachaeta* Hutchings, 1977 is only known by the type species and has only been found in the front of mangroves in Broken Bay, NSW and Moreton Bay, Queensland and, despite extensive sampling in these habitats along the east coast of Australia, no other material of this species has been found (Hutchings, pers., comm.).

### 3.4.3. Distribution of Terebelliforms with Depth

There is no generally accepted definition of the deep sea. One can consider depths below the euphotic layer (i.e., 300 m) as a natural upper limit to the deep sea. Overall, the deep-sea remains poorly explored outside of specific areas, such as cold seeps, hydrothermal vents, and organic falls. The typical lifestyle of terebelliforms also makes their capture unlikely by the gear typically used to sample the deep sea. In particular, species that live buried in the sediment or attached to rocks are often missed by dredges and beam trawls used by most recent general study programmes. This was clearly demonstrated by Gunton et al. [104] who studied the polychaete fauna from depths off the east coast of Australia (1000–4000 m), and while ampharetids were very well represented, with over 300 specimens belonging to more than six species, 10 specimens and 2 species of pectinariids were also present and described (*Petta investigatoris* Zhang, Hutchings and Kupriyanova, 2019 and *P. williamsonae* Zhang, Hutchings and Kupriyanova, 2019), and far fewer specimens of Terebellidae s.l. were collected, representing four genera but all in poor condition.

The deepest record among Terebelliformia comes from a species of ampharetid *Anobothrus auriculatus* Alalykina and Polyakova, 2020, found at 9584 m depth (Table 2). Ampharetids are well represented in the abyss and in different deep-sea habitats, with more than half of the known ampharetid species occurring below 500 m deep (Table 2). Several ampharetids are exclusively found in the deep-sea, in addition to some specialised representatives associated with chemosynthetic environments, such as some known species of the genera *Amage* (1 species., at cold seep), *Amphisamytha* (7 species., at cold seeps and hydrothermal vents), *Anobothrus* (3 species., at cold seeps and hydrothermal vent), *Decemunciger* (1 species., on decaying wood), *Endecamera* (1 species., on decaying wood), *Glyphanostomum* (2 species, at cold seep and sedimented hydrothermal vents), *Grassleia* (1 species at sedimented vents and cold seeps), *Paramytha* (2 species on decaying bones and sedimented hydrothermal vents), and *Pavelius* (3 species at cold seep and sedimented hydrothermal vents) [105].

The alvinellids are restricted to hydrothermal vents. All the species are exclusively found at hydrothermal vents in the Eastern and Western Pacific (Table 2), but a species was recently reported from vents in the Indian Ocean [7]. As a result, alvinellids are exclusively found at depths greater than 1500 m and can reach ~3600 m (Table 2).

Hydrothermal vents and cold seeps are also home to some terebellid species described recently (e.g., *Neoamphitrite hydrothermalis* Reuscher et al. 2012, and *Streblosoma kaia* Reuscher, Fiege and Wehe, 2012, for hydrothermal vents, and *Pista shizugawaensis* Nishi and Tanaka, 2006 for cold seeps; see [106] (Table 2). The Telothelepodidae, in contrast, so far have only been found at shallow depths, the deepest records (~91 m) for *Parathelepus anomalus* (Londoño-Mesa, 2009) and *Rhinothelepus mexicanus* (Glasby and Hutchings, 1986) (Table 2). In general, most polycirrids are found in intertidal to shallow water habitats, the deepest record being for *Polycirrus nonatoi* Carrerette and Nogueira, 2013, found from ~30–1900 m deep (Table 2). Trichobranchidae are also well represented in the deep sea, frequently by a large number of species (Table 2), despite sometimes being considered low in abundance, belonging to the genus *Terebellides*; the deepest records come from *Terebellides bulbosa* Schüller and Hutchings, 2012 and *T. gingko* Schüller and Hutchings, 2012, ~5200 m deep for both, from animals collected at the Brazil Basin [107] (Table 2). This genus is highly

speciose, with many endemic species, while others can have wide distributions, indicating dispersion over long distances [107]. Both Pectinariidae and Thelepodidae are much more diverse intertidally to ~100 m, but in both families a few species adapted to the deep sea have been described, the deepest records being *Streblosomma chilensis* (McIntosh, 1885), for thelepodids, registered at ~4000 m deep off Chile, and the pectinariid *Petta assimilis* McIntosh, 1885, found ~3000 m deep, off Crozet Islands (Table 2).

*3.5. Evolution of Methods Used to Describe Species*

Earlier taxonomists in the 18th and 19th centuries worked with very rudimentary optical instruments, sometimes only a little more than a magnifying glass, capable of low magnifications. Nevertheless, albeit with limited resources, those authors did an amazing job. Except for the Alvinellidae, Telothelepodidae and Thelepodidae, all other families, 35% of the genera and 23% of the currently valid species of Terebelliformia, were described in the 18th and 19th centuries (see above). Those descriptions are frequently criticized for their simplicity, but they reflect the state of knowledge at those times, when the authors considered enough to define species characters which, nowadays, frequently do not allow even for the recognition of the genus. Additionally, it is noteworthy how, in spite of these instruments, some of those earlier descriptions included minutely detailed drawings of chaetae, showing structures which can only be clearly seen under the SEM, a technology that was obviously developed much later.

A great improvement on taxonomists' instruments came in the 20th century, first with more powerful compound optical microscopes, with techniques such as phase contrast, allowing for a much better visualization of chaetal ornamentation, together with better software to capture the images directly from the microscopes and process them, largely replacing traditional line drawings (Figure 9).

Then, from the end of the 20th century through to today, Scanning Electron Microscopy (SEM) provides a much better view of the surface of microscopic structures, such as chaetal ornamentation (Figure 9); Transmission Electron Microscopy (TEM) brought information on cellular ultrastructure; confocal microscopy allowed us to peer deep into the tissues and highlight specific organs; molecular tools became available to distinguish the taxa at the genetic level (DNA and RNA), providing much more detailed and complete descriptions.

All this has greatly increased our knowledge on the diversity of terebelliforms, as for all other polychaetes, with many more morphological and molecular characters available to characterise the taxa, allowing for the recognition of complexes of cryptic species and alien species, for example, opposing the traditional view of species with wide distributions or even cosmopolitan [5,92].

Molecular data are extremely useful to delimit new species or even identify valid species, however it must be accompanied by voucher specimens and preferably be obtained from animals from the type localities of the species, in the case of those already described. Molecular data of misidentified species can generate much confusion. Additionally, in most cases, type species of the genera were not sequenced yet and cannot be included in the resulting phylogenies, compromising all the results obtained. Molecular studies on Terebelliformia so far have resulted in 222,406 sequences available for Alvinellidae in Genbank (mostly transcriptomics and phylogenetic markers), 1011 for Ampharetidae, 1298 for Pectinariidae, 2588 for Terebellidae s.l., and 1476 for Trichobranchidae, considering mitochondrial and nuclear gene markers (Figure 10) (Table 3).

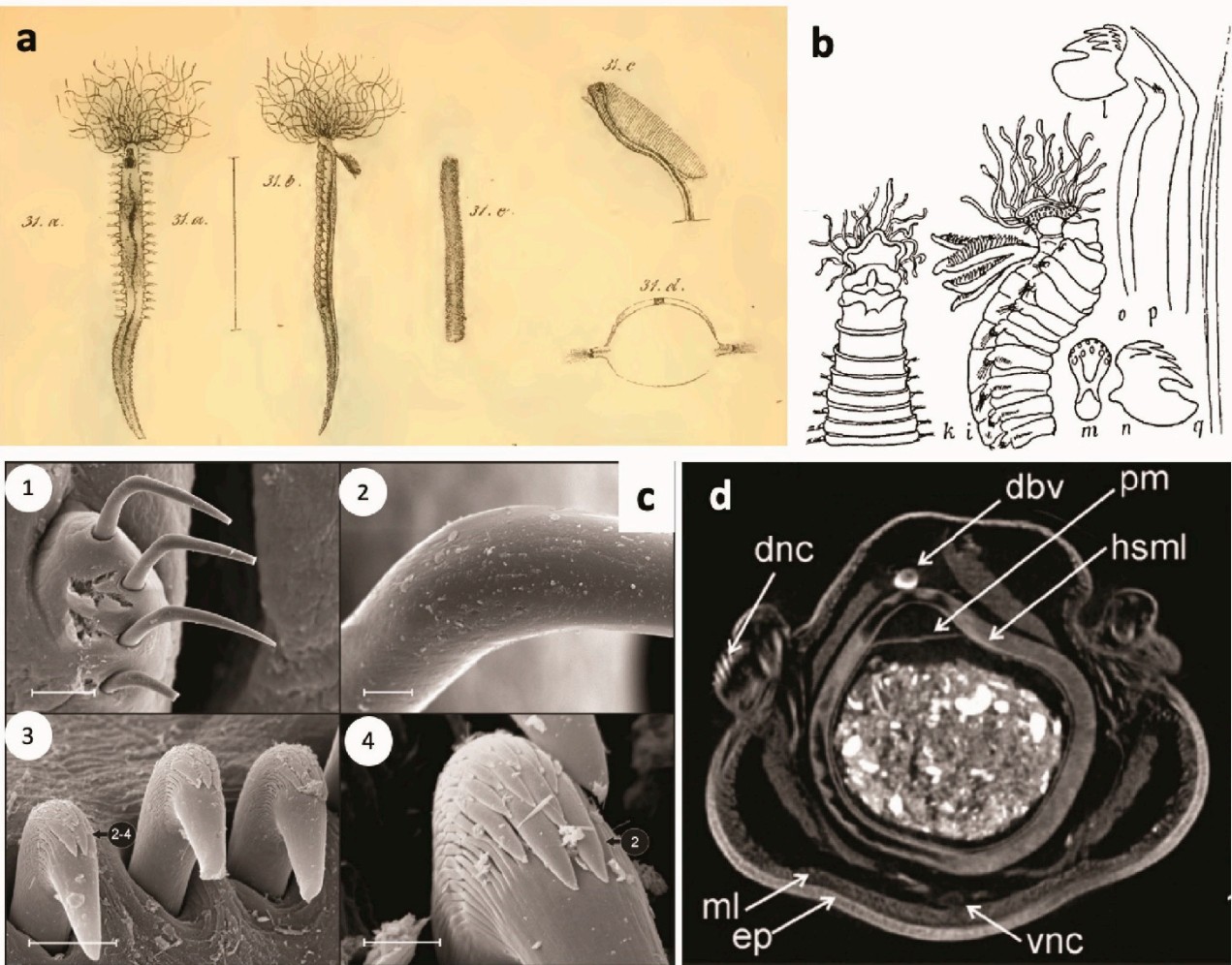

**Figure 9.** Evolution of appreciation of morphological traits important in taxonomy and techniques used, with *Terebellides stroemii* as an example. (**a**) Illustrations of the original description by Sars [108] (part of plate 13). (**b**) Later observations focused on optical microscopy of the chaetae (setae), here summarized by Fauvel [86]. Key: i side view of anterior part; k ventral view; l–n, uncini front and side view; o, thoracic ventral hook; p, geniculate chaeta; q, dorsal chaeta. (**c**) Scanning Electron Microscopy of geniculate chaetae (1) and detail of the bend (2), thoracic uncini (3) and detail of the teeth (4) (Parapar et al. [109]). (**d**) Micro-computed tomography (μCT) allows cross sections to look at the anatomical level. Here, section at the level of thoracic chaetiger 9. dnc, dorsal notochaetae; dbv, dorsal blood vessel; hsml, hind stomach muscle layer; ep, epidermis; ml, muscle layer; pm, peritrophic membrane; vnc, ventral nerve chord. (Modified after Parapar and Hutchings [110]). Scale bars a 50 μm, b 5 μm, c 10 μm, and d 3 μm.

Morphology-based polychaete taxonomy is largely based on external characters, particularly in Terebelliformia. Hessle [21] suggested a classification based on the structure of nephridia, however this requires dissection of the specimens, which is not feasible with museum material. However, a technique recently developed, computerized microtomography (μCT), scans the specimens and gives amazing 3D images of their internal anatomy (Figure 9), not causing any damage to the specimens, thus allowing type material to be examined. So far, few terebelliforms have been examined under the μCT, but as more are subjected to such analyses [111], additional morphological characters will certainly be found, increasing our knowledge on these animals.

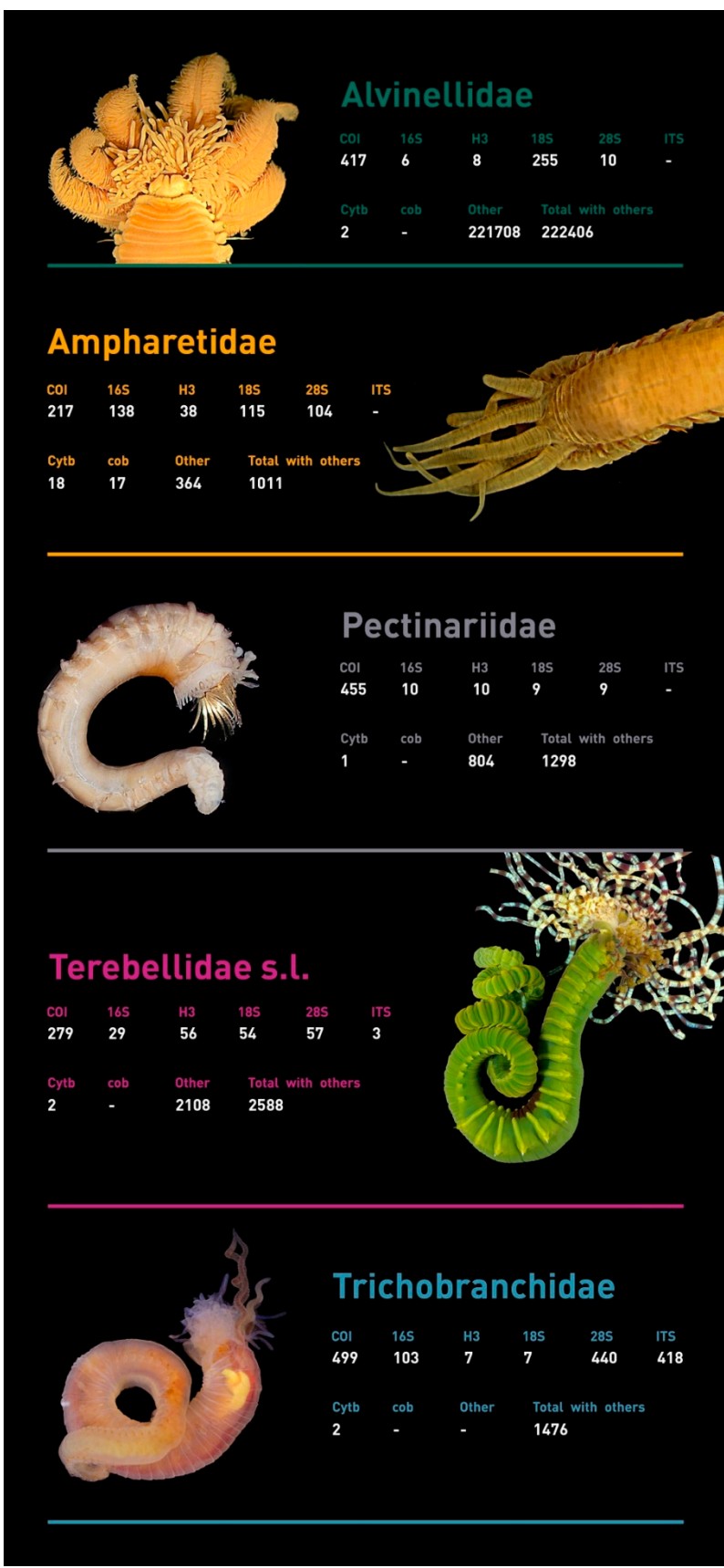

**Figure 10.** DNA sequences for Terebelliformia.

**Table 3.** Sequences available in Genbank for each main group of Terebelliformia.

| | COI | 16S | H3 | 18S | 28S | ITS | Cytb | cob | Other | Total | % Sequences with Voucher |
|---|---|---|---|---|---|---|---|---|---|---|---|
| Alvinellidae | 417 | 6 | 8 | 255 | 10 | 0 | 2 | 0 | 221,708 | 222,406 | 0.0 |
| Ampharetidae | 217 | 138 | 38 | 115 | 104 | 0 | 18 | 17 | 364 | 1011 | 53.9 |
| Pectinariidae | 455 | 10 | 10 | 9 | 9 | 0 | 1 | 0 | 804 | 1298 | 7.6 |
| Terebellidae s.l. | 279 | 29 | 56 | 54 | 57 | 3 | 2 | 0 | 2108 | 2588 | 19.7 |
| Trichobranchidae | 499 | 103 | 7 | 7 | 440 | 418 | 2 | 0 | 0 | 1476 | 793 |

*3.6. Knowledge Gaps and Challenges for the Future*

3.6.1. Poorly Known Regions of the World

As discussed above, some regions of the world have their local fauna of terebelliforms poorly known, as reflected by the low number of species originally described from those areas. In most cases, they correspond to poorly investigated areas of the world, such as the African coast (except for the Mediterranean part), Southern and Southeastern Asia, the western side of tropical America, corresponding to the Pacific Latin America shore, and all the Eastern Indo-Pacific realm, including the region from Hawaii and the Marshall Islands through Polynesia and the Mariana Islands to Easter Island. Those areas in most cases correspond to developing countries and/or with few institutions investigating invertebrate biodiversity. In some, however, the areas have been sampled and studied, but the identification of the specimens was made based on traditional monographs from other regions of the world, such as France [86,93], or South Africa [88], and resulting in many so called "cosmopolitan" species being recorded [92,112], whereas in fact they actually represent undescribed species. Even worse is that these names become incorporated into the ecological literature with no discussion as to the likelihood that a European species is present in China, for example (see [113]). With taxonomic studies of the fauna of Africa, India, China and other countries from SE Asia, and the Pacific side of America, the number of new species will certainly increase in the next decade or so and will mirror the tendency of the last decade (Figure 4).

Overall, the deep-sea also needs to be better explored, especially in areas that are not influenced by chemosynthetic local primary production. Better adapted gear for the sampling of sediment in which some species live may also be designed or adapted from other existing equipment.

Our knowledge of the deep-sea representatives of the terebelliforms, however, will soon expand as programmes are being conducted by many institutions around the world, such as the "Tropical Deep-Sea Benthos" series of cruises carried by the Museum National d'Histoire Naturelle, Paris, which have recently accessioned a large number of specimens to their collections. The use of tools, such as remotely operated vehicles (ROVs), has also allowed the targeted collection of deep-sea samples, and these are making their way to taxonomists around the world.

3.6.2. Species Complexes

Recently, with the rejection of cosmopolitanism [92] and the wide use of modern tools such as SEM imaging and molecular analysis, scientists have re-examined well-known species from well-known areas in Europe, resulting in the description of several cryptic species, as new to science. Consequently, the number of terebelliform species continues to increase, and many species previously considered widely distributed have become restricted to smaller areas. One of the best examples is *Terebellides stroemii*, reported from all around the world but almost certainly restricted to Norwegian waters [5]. These authors, using molecular data, showed the presence of more than 25 species in the Northeastern Atlantic alone, hidden behind this so-called "cosmopolitan" species. Parapar et al. [94] has just formally described five of these species identified by Nygren et al. [5]. By launching the Spaghetti project, Lavesque and collaborators are revising all French species of Terebellidae s.l. This project has allowed them to describe nine new species of Trichobranchidae [99],

three species of Thelepodidae [98] and eight species of Polycirridae [97] from French waters, an area historically well studied by early polychaetes workers (Audouin, Caullery, Fauvel, Gravier, Quatrefages, Rullier, Saint-Joseph, Savigny, etc.). A subsequent paper will document the diversity of Terebellidae from French waters (Lavesque et al. in prep.).

### 3.6.3. Taxonomic Issues Which Need to Be Resolved

Several genera of terebelliforms are monotypic, known only from the original descriptions, which do not include many characters important for the taxonomy of these groups, and type material is lost, damaged or cannot be located. In many cases, the material was collected by earlier expeditions and corresponds to species described, for example, by Grube, Müller, Lamarck, McIntosh, Chamberlin and Caullery. In some cases, the descriptions and illustrations are such that it is impossible to define the genus and, in these cases, they must be declared as *nomen dubium*, or indeterminable, at least until more material from the type of locality is collected and a neotype designated. Currently the genera *Paralanice* Caullery, 1944, *Opisthopista* Caullery, 1944 and *Spiroverma* Uchida, 1968, in the Terebellidae s.s., cannot be defined. Ebbe and Purschke [8] also list the monotypic genera, *Aryandes* Kinberg, 1866, *Rytocephalus* Quatrefages, 1866 and *Uschakovius* Laubier, 1973, as of doubtful affiliation. In some of the other cases, genera are not well known and Nogueira et al. [1] list those which could not be included in their phylogenetic study as the type material was either poorly preserved or too incomplete for scoring.

Another example is *Hadrachaeta* Hutchings, 1977. Although the type of locality has been extensively sampled through the years, since the original description, no additional specimens of *H. aspeta* Hutchings, 1977 have been obtained (Hutchings, pers. obs. [1]), and the type of material has been dissected several times, removing important diagnostic characters.

Another issue is whether some characters should be regarded as generic or species characters. These include the number of pairs of branchiae; in some genera, such as *Nicolea* Malmgren, 1866, they have two pairs, whereas in other genera the number of pairs is used to distinguish between species, such as in *Pista* Malmgren (2–3) and *Terebella* Linnaeus, 1767 (2–3, although the segment on which they occur can vary).

In *Pista*, the type of branching of the branchiae is a specific character. However, the genus *Pistella* which has only one pair of branchiae resembling some *Pista* species has recently been synonymised with *Pista* by Jirkov and Leontovich [63] but lacks the long-handled uncini characteristic of *Pista*. This is complicated by the type species of *Pista* (*Amphitrite cristata* Müller 1776) which was described as having one pair of branchiae, while Malmgren who erected the genus *Pista* and designated *P. cristata* (Müller 1776) as the type species, stated it has two pairs of branchiae, and no type material exists. However, this synonymy has between *Pista* and *Pistella* has not been accepted by other workers, and Hutchings et al. [2] record 76 species currently assigned to *Pista*, whereas the genus *Pistella* has four species.

Another issue which needs to be resolved is the development of long-handled uncini on thoracic neuropodia, which occur in several terebellid genera and their actual structure. Jirkov and Leontovich [63] have also suggested that all genera with long-handled uncini be synonymised with *Axionice* and that such structures are specific and not generic characters. This hypothesis has not been accepted but highlights the need for more developmental studies to actually study the development of the branchiae and the chaetae as the larvae settle and become juveniles. Similarly, the development and homologies of the peristomium and prostomium needs to be carefully investigated by developmental studies. Finally, the development of the anterior lateral lobes needs to be examined in detail, as their shape, orientation and the segment on which they occur appear to be very useful specific characters in many genera, although Jirkov and Leontovich [63] have suggested that all genera with large lateral lobes be synonymised, although they do not explain why this should happen.

A final issue concerns the genus *Pseudothelepus* Augener, 1918. Augener described this genus for *P. nyanganus* Augener, 1918, from the Tropical Atlantic coast of Africa. Later, Hartman [74] incorrectly synonymised *P. nyanganus* with *Sabellides oligocirra* Schmarda, 1861, described from the Caribbean, keeping the validity of the genus *Pseudothelepus* and changing the type species to *P. oligocirrus*. Unaware that the name was preoccupied, Hutchings [26] described an unusual thelepodid from Houtman Abrolhos Islands, Western Australia, as a new genus and species, which she named *Pseudothelepus binara* Hutchings, 1997. One of us (J.M.M.N.) examined the type of material of the three species and verified that all three are separate, valid species, rejecting the synonymy between *P. nyanganus* and *S. oligocirra*. However, both *P. nyanganus* and *S. oligocirra* are species of *Streblosoma*, and therefore *Pseudothelepus* is not valid. *Pseudothelepus binara*, in contrast, is a very different species, which justifies the erection of a new genus, since the original name is preoccupied, although that still requires phylogenetic confirmation.

So, in summary, not only will new species continue to be described around the world, but a more robust discussion needs to be had on the way in which generic and specific characters are defined, as well as better descriptions of those type species, which are currently inadequate. Ideally these descriptions will be based on neotypes and ideally with associated molecular data.

## 4. Discussion

As our taxonomic knowledge of this large group of polychaetes (both in terms of diversity and abundance) continues to increase, we need to develop online resources to make these data widely available to the wider biological community. Currently, online keys to the families are being developed and will be uploaded when completed (Kupriyanova et al. in prep.), which include all annelid families and genera as well as Australian species. Similar guides need to be developed for other parts of the world and the views that old monographs such as [86,93] and [88] should not be used and instead retained as historical documents [112] should become widely accepted.

An initiative in Australia could be developed elsewhere. The Atlas of Living Australia (https://www.ala.org.au/) is regularly updated by all the State natural history museums who upload their registered collection onto ALA. These data are all specimen based and you can interrogate the data and download distribution maps, as shown in Figure 11, which shows all the terebellid s.l. data from around Australia and indicates the number of species recorded around the coast which have all been checked by Hutchings and her colleagues. Similar analyses could be carried out in other parts of the world, but one needs to check the validity of the original identifications.

For example, if such data from terebellids as a selective deposit feeding group are combined with other polychaete families, which are filter feeders, such as the sabellids/serpulids, and opportunistic feeders, such as nereidids, for which the taxonomic data are good, one would be able to characterise benthic communities. Such data would be invaluable when developing zoning plans for marine national parks, which currently are often based on physical parameters, such as depth, sediment type, surrogates, such as seagrass beds, sponge gardens, coral reefs, and with limited biological data, such as fisheries data. Yet, the benthic communities dominate these parks and play a crucial role in the marine ecosystem and are barely considered. The sort of data which can be extracted from ALA could provide valuable data to improve the representativeness of marine parks and help develop monitoring programs to ensure that such plans are effectively conserving their biodiversity. Critically important is the fact that climate change is impacting our marine communities.

If we, taxonomists, can provide this sort of data to ecologists, marine managers, this may enhance our ability to attract funds to continue our research and to facilitate the training and mentoring of the next generation of taxonomists.

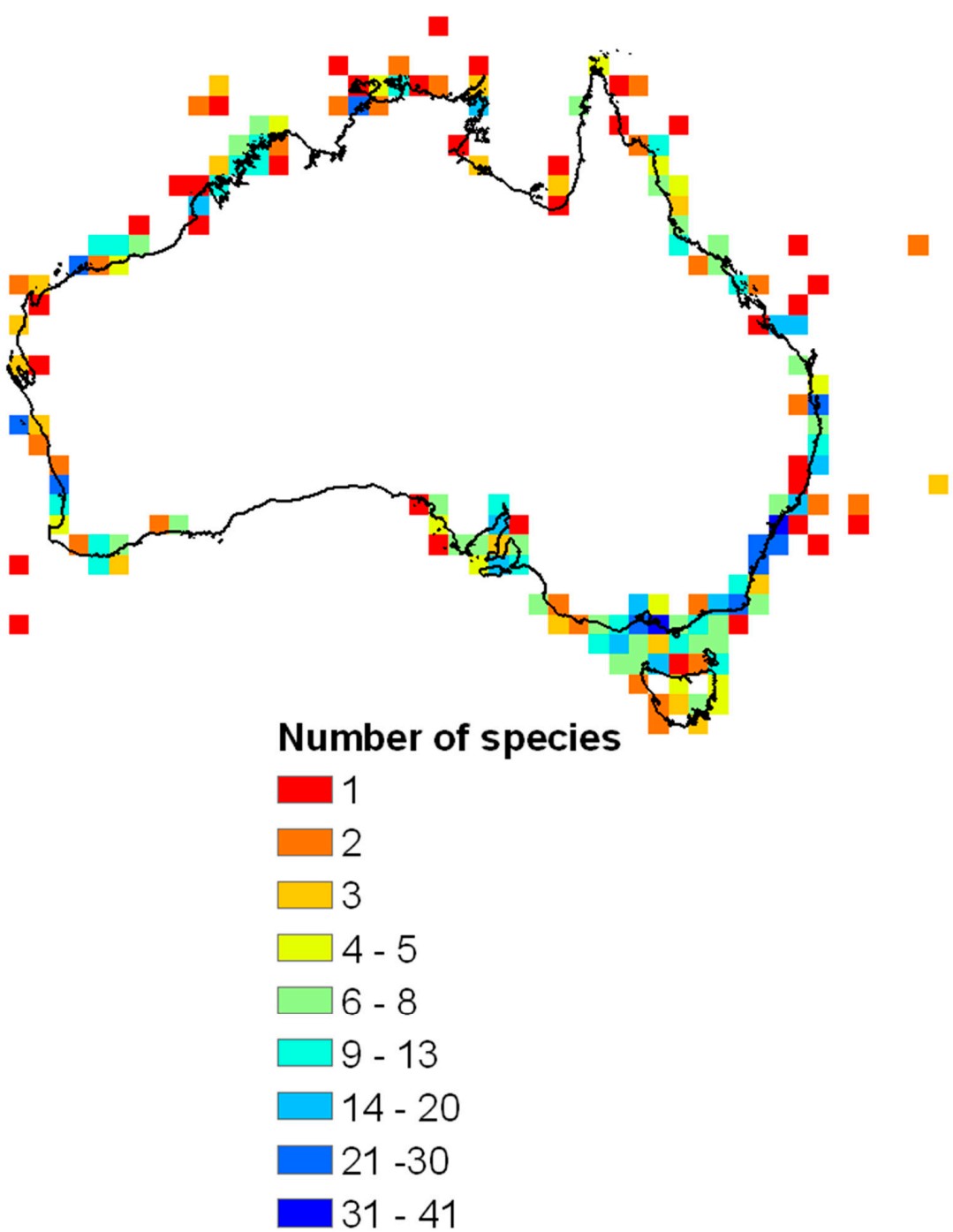

**Figure 11.** The distribution of terebellids ss. species around Australia based on data from ALA.

**Supplementary Materials:** The following are available online at https://www.mdpi.com/1424-281 8/13/2/60/s1.

**Author Contributions:** Conceptualization, P.H., J.M.M.N. and N.L.; validation, P.H., J.M.M.N., O.C., N.L. and S.H.; writing—original draft preparation P.H., J.M.M.N., O.C., N.L. and S.H.; writing—review and editing, P.H., J.M.M.N., O.C., N.L. and S.H.; visualization, J.M.M.N., O.C., N.L. and S.H.; coordination; P.H., J.M.M.N., O.C., N.L. and S.H.; funding acquisition, P.H. All authors have read and agreed to the published version of the manuscript.

**Funding:** O.C. receives a post-doctoral fellowship from FUSP and Shell Brasil; J.M.M.N. receives a productivity grant from CNPq, "Conselho Nacional de Desenvolvimento Científico e Tecnológico,

Brazil, level 2; N.L. has received financial support from the French State in the frame of the "Investments for the future" Programme IdEx Bordeaux, reference ANR-10-IDEX-03-02.

**Acknowledgments:** Alisson Ricardo da Silva (Museu da Casa Brasileira, Brazil) helped with the figures.

**Conflicts of Interest:** The authors declare no conflict of interest. The funders had no role in the design of the study; in the collection, analyses, or interpretation of data; in the writing of the manuscript, or in the decision to publish the results.

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
