# Peer review of "The Terebelliformia-Recent Developments and Future Directions"

_diversity, doi:10.3390/d13020060_

Round 1

Reviewer 1 Report

This is a good review based in the deep knowledge of the different authors regarding this Polychaeta group. I only have some minor comments

Introduction

Ln29-37 – maybe this paragraph should be after Ln60

Ln 64, aim 6 –what is discussed further ahead (discussion) is really reduced maybe remove this goal or extend the discussion

Missing some background information about depth range, latitude and habitats for instance – globally just to introduce the thematic

Material and Methods

Good historical review. Please check Figure 6 – I only count 36 species between 1760 and 1850… the authors mentioned 45 in the text (ln 110) please confirm the data represented

Ln 351 – could not find this reference

3.4.1 Maybe add a table with functional traits for different Families

Ln537 – reads as Introduction

3.4.3 Why not merge with 3.4.2

Table 2 –I guess there is some species overlap or the number is restricted to a specific region? Is not clear

Figure 10 – Maybe add the number of species sequenced… X sequences could be of the same species. It would be interesting to have that information -  and analyse the effort of sequencing for the different Families

3.7 – I guess it is also dependent where the taxonomist is located

Fig 11 – it works as an example but is really specific to Australia. Is not possible to that exercise for the correspondent marine ecoregions

References

6 – please correct

Author Response

Thankyou for a positive review

we decided against moving Ln 29-37 to later

We have retained Line 64  aim 6 -  and feel that this is included in the Discussion

Material and Methods we have double checked all the nos of species, found some mistakes and corrected text and the relevent figure

we have added some Supplementary Table of species 

Line 351- added

3.4.1 we have not added a table of functional traits for different families, as most would all be similar

Table 2 little overlap of species between regions

Fig 10- not useful to add nos of species sequenced as many incorrect species identiifications 

Fig 11- just using as an example--  to provide a map of other regions would be difficult as many records in data bases like OBis are incorrect 

we have corrected the format of refs

Reviewer 2 Report

This manuscript summarises the actual knowledge on several annelid families reunited under the Terebelliformia. I think is an excellent example of how to put together a synthesis about systematics, diversity and ecology while highlighting gaps on knowledge and explicitly acknowledging contributions of early authors that did so much while lacking the equipment and some techniques available to us now. I also particularly liked how figures and tables were organised. I did not find any major omissions or relevant issues. Therefore, I recommend this paper for publication after fixing some minor mistakes and typos that I detailed below and in the attached pdf version the MS. My sincere congratulations to the authors for such a fine paper.

Minor comments:

Introduction, Page 1, Line 29: Stating who erected Terebelliformia or some succinct information about the history of this taxon?

References in text: there are both numbers and year of publication.

Figure 1, 2 and others: There are pictures of several terebelliform species but it is not stated to what family do they belong. What about including an abbreviation for each family and placing it after the species name? e.g. PE, AL, AM, PO

Figure 1, 2, 3 legends: "Holotype" and "Topotype" are written with first letter in capitals in Figure 1 while in others are not and "paratype" is always written in small caps. It also would be interesting to highlight which specimens are stained (Fig. 1c), if they were alive when the picture was taken, etc. Staining is indicated for instance in Figure 5 legend.

Figure 5 legend: Is "lazo-wasemi" properly written according to the ICZN rules?

Please carefully check the format in the references' list. I just highlighted some minor mistakes. DOI numbers are indicated only for some references.

Author Response

Again we should like to thank the reviewer

we have accepted all the changes in the manuscript re spacing and minor comments and refs and added the additional refs